# Thermophoretic glycan profiling of extracellular vesicles for triple-negative breast cancer management

Yike Li[1,2,8], Shaohua Zhang [3,8], Chao Liu [1,4] ✉, Jinqi Deng[1,4], Fei Tian[1,4], Qiang Feng [1], Lili Qin[3], Lixiao Bai[3], Ting Fu[5,6], Liqin Zhang[2] ✉, Yuguang Wang[7] ✉ & Jiashu Sun [1,4] ✉

Triple-negative breast cancer (TNBC) is a highly metastatic and heterogeneous type of breast cancer with poor outcomes. Precise, non-invasive methods for diagnosis, monitoring and prognosis of TNBC are particularly challenging due to a paucity of TNBC biomarkers. Glycans on extracellular vesicles (EVs) hold the promise as valuable biomarkers, but conventional methods for glycan analysis are not feasible in clinical practice. Here, we report that a lectin-based thermophoretic assay (EVLET) streamlines vibrating membrane filtration (VMF) and thermophoretic amplification, allowing for rapid, sensitive, selective and cost-effective EV glycan profiling in TNBC plasma. A pilot cohort study shows that the EV glycan signature reaches 91% accuracy for TNBC detection and 96% accuracy for longitudinal monitoring of TNBC therapeutic response. Moreover, we demonstrate the potential of EV glycan signature for predicting TNBC progression. Our EVLET system lays the foundation for non-invasive cancer management by EV glycans.

Breast cancer (BC) is the most commonly diagnosed cancer worldwide and the second leading cause of cancer death in women[1]. Triple-negative breast cancer (TNBC), as clinically defined by the lack of expression of estrogen receptor (ER), progesterone receptor (PR) and human epidermal growth factor receptor 2 (HER2)[2], is a highly aggressive subtype of BC with poor prognosis[3,4]. TNBC is currently diagnosed based on the combination of imaging (mammography X-ray photographic examination) and immunohistochemistry, while monitoring treatment response in TNBC relies on the frequent use of ultrasonography and magnetic resonance imaging[5,6]. These approaches are operator dependent, high cost and inaccessible to the general public. Although blood-based liquid biopsy shows immense potential for non-invasive cancer diagnosis and monitoring, a paucity of biomarkers in TNBC limits the diagnostic efficiency of blood test.

EVs are a heterogeneous group of lipid bilayer-enclosed nanovesicles secreted by most cell types, and have been widely explored as emerging circulating biomarkers for liquid biopsy[7–18]. In addition to carrying a myriad of molecular cargos inherent from parental cells, tumor-derived EVs are also heavily enriched in glycoconjugates which play crucial roles in different steps of tumor progression[19–25]. Despite the promise of EV glycans as non-invasive biomarkers for cancer, their clinical applications are hindered by the complexity of glycan structures, the

[1]Beijing Engineering Research Center for BioNanotechnology, CAS Key Laboratory of Standardization and Measurement for Nanotechnology, National Center for Nanoscience and Technology, Beijing 100190, China. [2]State Key Laboratory of Natural and Biomimetic Drugs, School of Pharmaceutical Sciences, Peking University, Beijing 100191, China. [3]Department of Oncology, the Fifth Medical Center of PLA General Hospital, Beijing 100071, China. [4]School of Future Technology, University of Chinese Academy of Sciences, Beijing 100049, China. [5]Hangzhou Institute of Medicine (HIM), Chinese Academy of Sciences, Key Laboratory of Zhejiang Province for Aptamers and Theranostics, Hangzhou, Zhejiang 310022, China. [6]Jiangsu Union Institute of Translational Medicine, Zhongdi Biotechnology Co., Ltd, Nanjing, Jiangsu 211500, China. [7]Department of General Dentistry II, Peking University School and Hospital of Stomatology & National Center for Stomatology & National Clinical Research Center for Oral Diseases & National Engineering Research Center of Oral Biomaterials and Digital Medical Devices, Beijing 100081, China. [8]These authors contributed equally: Yike Li, Shaohua Zhang. ✉e-mail: liuc@nanoctr.cn; lqzhang@hsc.pku.edu.cn; wangyuguang@bjmu.edu.cn; sunjs@nanoctr.cn

heterogeneity of EVs in body fluids and the lack of practical analytical methods. Mass spectrometry (MS) for profiling EV glycans requires expensive instrumentation, multiple procedures of EV isolation and glycan release, and the experienced operators[26,27]. Lectin microarray is another platform for EV glycan analysis by immobilizing a variety of carbohydrate-binding proteins (also known as lectins) onto a glass substrate for global glycomic profiling[28,29]. In spite of its feasibility, the sensitivity of lectin microarray could be compromised due to the diffusion-limited mass transport of EVs[30]. Recent development of microfluidic platforms enables the analysis of EV glycans in clinical specimens by transducing EV glycan signatures into magnetic signals, or using intra-assembly spatial distribution of nanoparticles to generate differential fluorescence signals[20,31]. These microfluidic methods have great potential for clinical detection of EVs. However, the fabrication of microfluidic sensors may require access to clean rooms and specialized skills.

To address the current challenges, recent work has devised a rapid, sensitive, low-cost and aptamer-based thermophoretic assay to profile tumor-related EV proteins using a small volume of serum or plasma samples with minimal operator involvement[32–35]. Nevertheless, the adaption of thermophoretic assays to detect EV glycans in TNBC patients requires several improvements. These include (i) identification of TNBC-associated glycans based on bioinformatics tools and data resources, (ii) effective removal of interfering lipoproteins, soluble glycoproteins and other substances that affect the specificity for detecting EV glycans, and (iii) sensitive detection of EV glycans and clinical validation of EV glycan signature for TNBC management. Here, we develop a lectin-based thermophoretic assay termed as EVLET for rapid, sensitive and selective analysis of EV glycan profiles in plasma samples within 100 min. We show that the EV glycan signature as the weighted sum of three lectins offers high accuracy for diagnosis, monitoring and prognosis of TNBC in 135 plasma samples.

## Results

### EVLET for sensitive and specific detection of EV glycans

The overall workflow of EVLET system is summarized in Fig. 1, which streamlines an automatic vibrating membrane filtration system (VMF) and a thermophoretic assay. First, 20 μL of 10-fold diluted plasma was incubated with FITC-conjugated lectins to specifically label TNBC-associated glycans on EV surfaces. The sample was processed by VMF for automatic isolation and purification of lectin-bound EVs by removing unbound lectins, soluble glycoproteins and lipoproteins (Supplementary Figs. 1–3). The isolated EVs were then transferred to a 400 μm high microchamber (7 mm in diameter) with a sapphire substrate, followed by thermophoretic accumulation of EVs to amplify the fluorescence signal of lectin-bound EVs. Using a panel of lectins, the obtained EV glycan signature was applied for TNBC diagnosis, treatment response monitoring and prognosis of TNBC patients.

### Identification of TNBC-associated glycans

To select TNBC-associated glycans, we first analyzed the mRNA expression levels in TNBC, other subtypes of BC and normal tissues from the Cancer Genome Atlas Breast Invasive Carcinoma (TCGA-BRCA) database (Fig. 2a). Among 17394 mRNAs available in the TCGA-BRCA cohort ($n = 178$ for TNBC, $n = 862$ for other subtypes of BC and $n = 113$ for adjacent normal tissues), 5525 mRNAs were found to be differentially expressed (false discovery rate < 0.001, |log$_2$ fold change | > 1) between TNBC and non-TNBC (other BC subtypes and normal tissues) by differential expression analysis (DEA) (Supplementary Fig. 4). Based on the GlycoGene database, 48 mRNAs were annotated as the mRNAs encoding glycosyltransferases or glycosidases that are involved in the biosynthesis of glycans[36]. Linear discriminant analysis (LDA) of 5525 TNBC-associated mRNAs or 48 glycosylation-related mRNAs revealed an overall accuracy of 98.7% or 92.9% for classification of TNBC, other BC subtypes and normal tissues in the TCGA-BRCA cohort ($n = 1153$, Fig. 2b), demonstrating the potential of glycan profiles for TNBC diagnosis. The t-distributed stochastic neighbor embedding (t-SNE) plot also confirmed that the set of 48 mRNAs was capable of TNBC discrimination (Supplementary Fig. 5). These glycosylation-related mRNAs are associated with diverse glycans including mannose, galactose, N-acetylgalactosamine, glucose, N-acetylglucosamine, sialic acid, fucose, glucuronic acid and lactose, which are regarded as

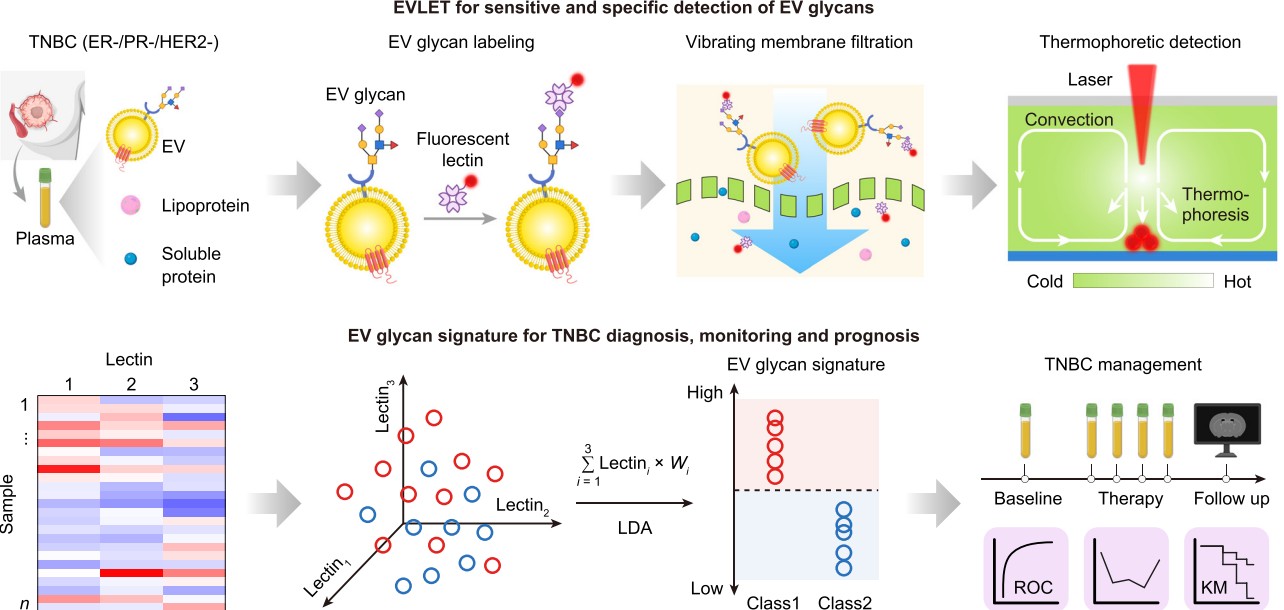

**Fig. 1 | Schematic of the EVLET system for sensitive and specific detection of EV glycans.** Plasma EVs were incubated with FITC-conjugated lectins for specifically binding TNBC-associated glycans on EV surfaces. A customized VMF was used for automatic, rapid, high-performance isolation and purification of lectin-labeled EVs by removing unbound lectins, soluble glycoproteins and lipoproteins.

A thermophoretic assay was applied to accumulate lectin-labeled EVs for an amplified fluorescence signal relative to specific EV glycans. An EV glycan signature was identified by machine learning algorithms for TNBC management. LDA indicates linear discriminant analysis. Created with BioRender.com.

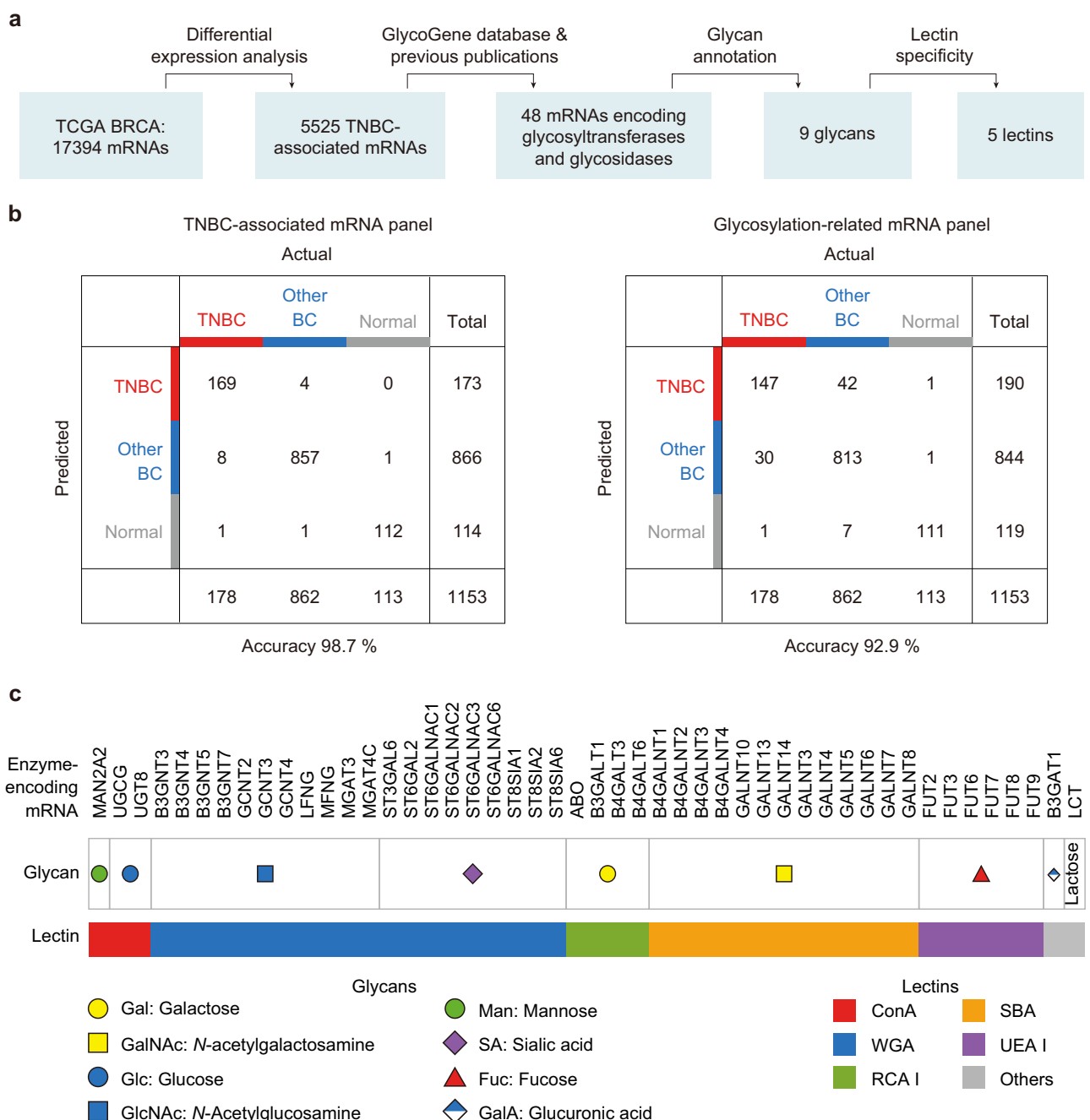

**Fig. 2 | Identification of TNBC-associated glycans and the corresponding lectins. a** Workflow for the identification of TNBC-associated glycans using TCGA-BRCA and GlycoGene, and the selection of lectins for glycan recognition. **b** LDA classification of TNBC, other BC subtypes and normal tissues based on 5525 TNBC-associated mRNAs or 48 glycosylation-related mRNAs. **c** TNBC-associated glycan epitopes and a panel of 5 lectins for detecting most of glycans.

TNBC-associated glycans (Fig. 2c)[36]. We further identify a panel of lectins (ConA, WGA, RCA I, SBA and UEA I) that can bind to most of these glycans (Supplementary Table 1)[28,31].

### Design of EVLET

EVLET integrated a customized VMF for EV isolation and a thermophoretic assay for EV glycan detection. We first characterized the performance of VMF for automatic filtration, washing and retrieval of EVs secreted by TNBC-derived MDA-MB-231 cell line (Fig. 3a and Supplementary Fig. 1). VMF consisted of a nanoporous anodic aluminum oxide (AAO) membrane and a high-frequency oscillator (6 kHz, 25 V). The oscillator-enabled vibration of AAO membrane periodically lifted up the MDA-MB-231 EVs to prevent membrane fouling[37]. Despite the

nominal pore size of 20 nm, scanning electron microscopy (SEM) characterization indicated that ~20% of pores on AAO membrane were in the size range of 40–50 nm (Supplementary Fig. 3). Given that the majority of cell line derived-EVs were larger than 50 nm, MDA-MB-231 EVs could be retained on AAO membrane after VMF (Supplementary Table 2). As seen in SEM images, the isolated MDA-MB-231 EVs remained round-shaped with no detectable damage, while EVs smaller than 100 nm were also observed (Fig. 3b and Supplementary Fig. 6). The wide-field transmission electron microscope (TEM) and nanoparticle tracking analysis (NTA) further confirmed that the size distributions of MDA-MB-231 EVs were almost consistent before and after VMF (Supplementary Figs. 7–10). The percentage of MDA-MB-231 EVs smaller than 100 nm was slightly decreased from 27% to 24% after VMF.

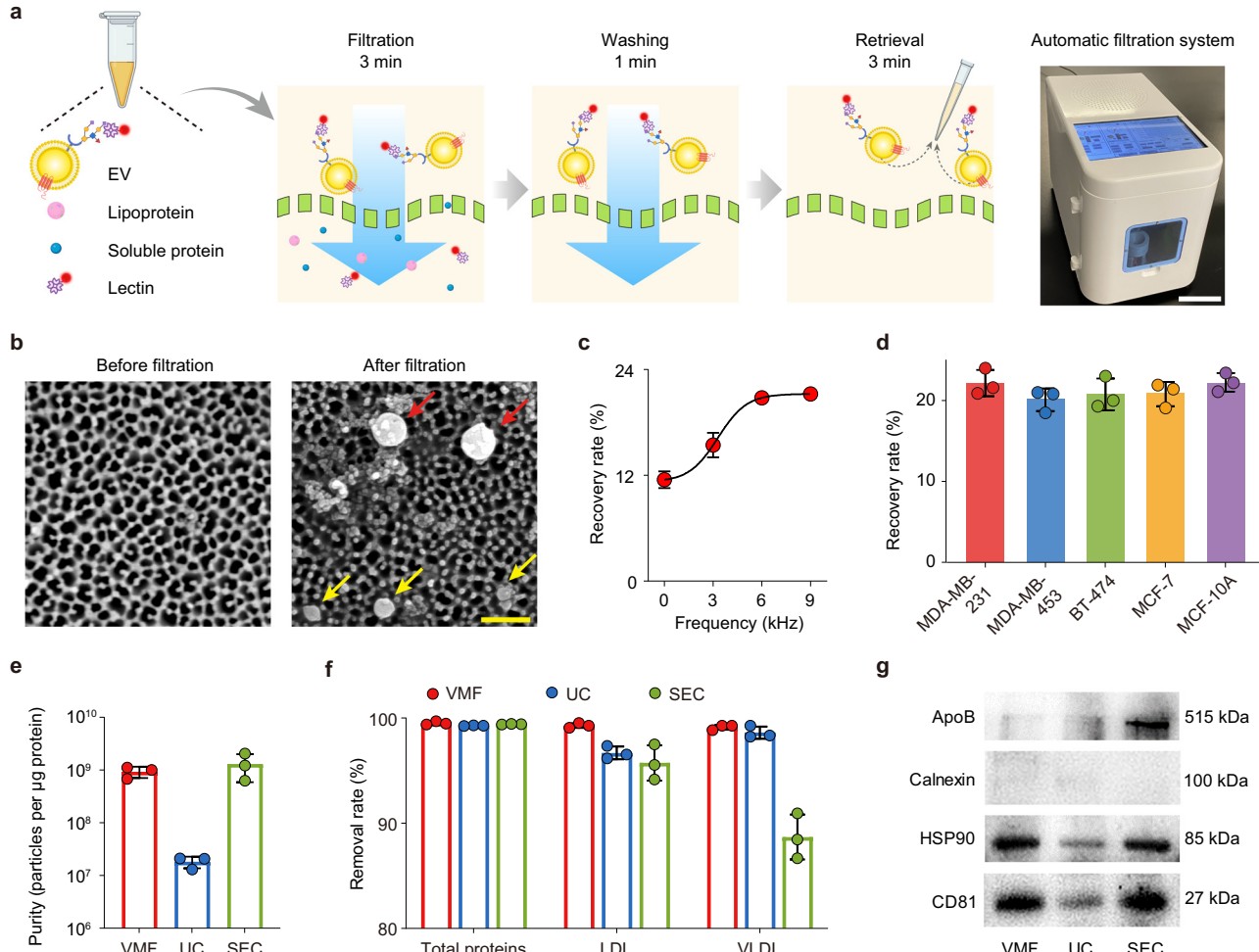

**Fig. 3 | VMF to isolate and purify EVs derived from BC cells and EVs from plasma samples. a** Schematic illustration of VMF for isolation and purification of EVs (left). Photograph of VMF system (right). Created with BioRender.com. Scale bar, 10 cm. **b** SEM images of the AAO membrane before and after VMF. The isolated EVs marked by yellow arrows (smaller than 100 nm) and red arrows (larger than 100 nm) remained round-shaped with no detectable damage. The representative images are shown from three independent repeats. Scale bar, 200 nm. **c** EV recovery rate by VMF at different vibration frequencies ($n = 3$ samples for each

frequency). **d** Recovery rate of EVs from 5 different breast cell lines by VMF ($n = 3$ samples for each EV type). **e** Comparison of purity of EVs isolated from plasma samples by VMF, UC, or SEC ($n = 3$ samples for each method). **f** Comparison of removal rates of total proteins, LDL and VLDL in plasma samples using VMF, UC, or SEC ($n = 3$ samples for each type of contaminant). **g** Western blot analysis of recovered samples from plasma by VMF, UC, or SEC. The samples were loaded with equal protein amounts. Error bars represent the mean ± s.d. in (**c–f**). Source data are provided as a Source Data file.

The recovery rate of MDA-MB-231 EVs by VMF (6 kHz) was 22.2%, 1.9-fold higher than that without vibration (0 kHz, Fig. 3c). VMF showed similar recovery rates ranged from 20.1%–22.2% for EVs from 4 other breast cell lines (MDA-MB-453, MCF-7, BT-474 and MCF-10A), with a low coefficient of variation of 5.2%–9.5% (Fig. 3d and Supplementary Fig. 8).

We further applied VMF to isolate and purify EVs from plasma samples that contain large amounts of circulating proteins, low-density lipoproteins (LDL, 18–25 nm) and very low-density lipoproteins (VLDL, 30–80 nm with over 90% of VLDL smaller than 50 nm)[38,39] (Supplementary Figs. 11, 12). Benefiting from the large pores (40–50 nm) of AAO membranes, 99.5% of proteins, > 99.1% of LDL and > 98.9% of VLDL in plasma could be removed by VMF (details in methods, Supplementary Table 3). In addition, a comprehensive list of assays suggested by the MISEV-2018 document was carried out for determining the recovery and purity (the ratio of particle count to protein content) of EVs isolated by VMF and other methods[40–44]. As summarized in Supplementary Table 3, VMF yielded a recovery rate (22.1%) that was substantially higher than ultracentrifugation (UC, 1%), while being lower than size exclusion chromatography (SEC, 39.3%) and tangential flow filtration (TFF, 60–80%). Both VMF and SEC

exhibited the high purity ($9.3 \times 10^8$ particles per µg protein for VMF and $1.3 \times 10^9$ for SEC), which was better than UC ($1.8 \times 10^7$) and TFF ($10^7–1.2 \times 10^8$) (Fig. 3e). The removal rates of total proteins, LDL and VLDL by VMF were higher than UC and SEC (Fig. 3f and Supplementary Table 3). Western blotting (WB) of plasma EVs isolated by VMF showed the enrichment of EV markers, CD81 and HSP90, and the absence of non-EV markers, calnexin and ApoB, verifying the high-purity of plasma EVs after VMF (Fig. 3g). Compared to the other three methods, VMF is compatible for small sample volumes (down to 2 µL of plasma) and is fully automatic with a total processing time around 10 min (Supplementary Table 3), providing a powerful tool for EV isolation and purification.

Next, we adopted a thermophoretic assay for sensitive detection of EV glycans that were labeled by fluorescent lectins (Fig. 4). As unbound lectins may affect the detection accuracy, we first assessed the capability of VMF to remove free FITC-ConA without the presence of EVs. Due to its small size, 99.9% of FITC-ConA could be filtered out by VMF (Supplementary Fig. 13). To determine the percentage of lectin-conjugated EVs derived from different cell lines, we used FITC-ConA to label EV glycans and DiD (a lipophilic dye for EV membranes) to stain all EVs. Fluorescence analysis revealed that 25.8% of MDA-MB-

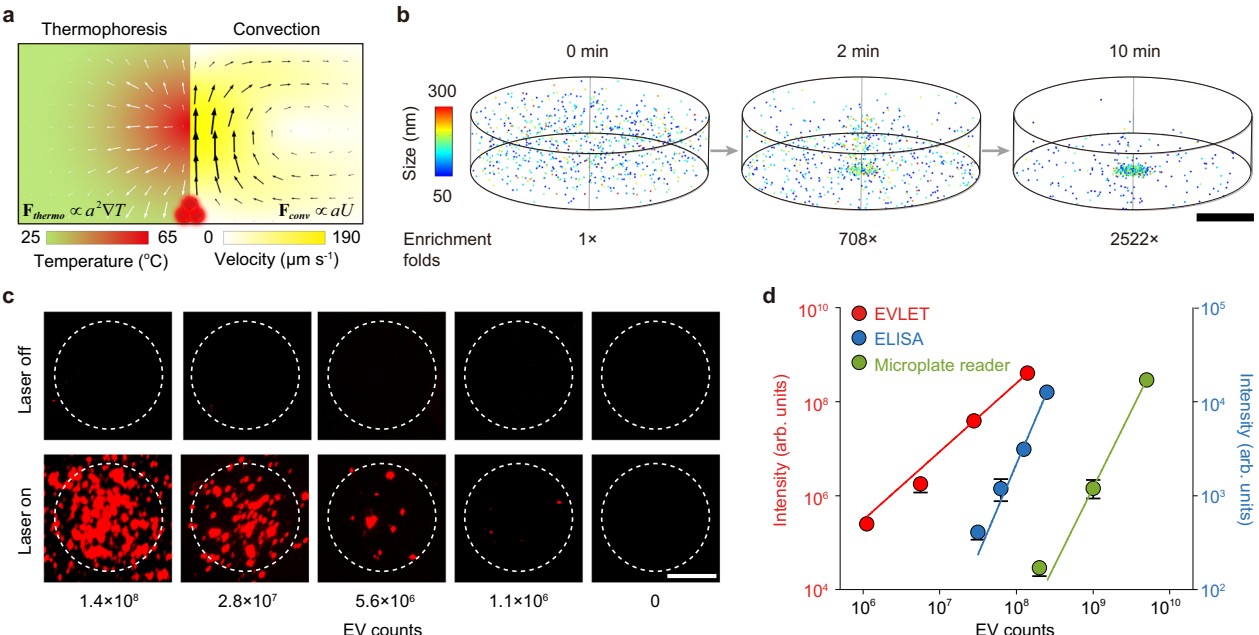

**Fig. 4 | Thermophoretic detection of EV glycans by EVLET. a** Mechanism of size-dependent accumulation of EVs under the interplay of thermophoresis and convection induced by the laser irradiation-generated temperature gradient ($\nabla T$). **b** Numerical simulation of the spatial distribution of EVs at different time points. Scale bar, 400 μm. **c** Fluorescence images of MDA-MB-231 EVs with various counts per assay detected by EVLET using ConA. The representative images are shown from three independent repeats. Scale bar, 50 μm. **d** Sensitivity of EVLET (red, the left axis), lectin-based ELISA (blue, the right axis) and direct fluorescence measurement by microplate reader (green, the right axis) for detecting glycans on MDA-MB-231 EVs using ConA ($n = 3$ samples for each EV count). Error bars represent the mean ± s.d. in (**d**). Source data are provided as a Source Data file.

231 EVs (TNBC cell line-derived EVs) and 6.6% of MCF-10A EVs (benign breast cell line-derived EVs) had DiD-FITC colocalization. In addition, MDA-MB-231 EVs exhibited a higher averaged FITC intensity than MCF-10A EVs (1.2 folds, $p = 0.035$) (Supplementary Fig. 14). These results suggested an elevated expression of glycans on TNBC-derived EVs. To improve the sensitivity of EV glycan detection, lectin-labeled EVs after VMF were loaded into a small microchamber and subjected to a 10-min local laser irradiation (1480 nm, 150 mW). The laser-induced radial temperature gradient between the heated core (up to 65 °C) and cold surrounding (25 °C) manifested two primary effects, thermophoresis that drove EVs away from the heated core (proportional to $a^2$, $a$ is the particle diameter), and convection flow for circulating EVs by exerting a drag force proportional to $a$ (Fig. 4a). The counterbalance between the two effects resulted in an over 2500-fold enrichment of MDA-MB-231 EVs (50–300 nm in size based on NTA characterization) at the chamber bottom, as indicated by simulation results in Fig. 4b. Notably, using a sapphire substrate with a high thermal conductivity (35 W m⁻¹ K⁻¹), the temperature around the bottom of microchamber was below 28 °C, mitigating the impact of heat on EV glycan analysis.

Experimental study confirmed that thermophoretic accumulation of FITC-ConA-labeled MDA-MB-231 EVs produced an amplified fluorescence signal in a confined region (100 μm in diameter), the intensity of which was linearly correlated with the EV concentration ($1.1 \times 10^6$ to $1.4 \times 10^8$ MDA-MB-231 EVs per assay, Fig. 4c, d). The limit of detection (LoD) was $4.1 \times 10^5$ EVs by EVLET, 2 orders of magnitude lower than that of lectin-based ELISA ($3.8 \times 10^7$ EVs) or direct fluorescence measurement by microplate reader ($1.1 \times 10^8$ EVs, Fig. 4d). After peptide-N-glycosidase F (PNGase F) treatment, the fluorescence intensity of ConA-labeled MDA-MB-231 EVs was reduced by 36% due to the cleavage of N-linked mannose ($p = 0.0024$, Supplementary Fig. 15). We further examined the impact of soluble proteins such as CA 125 and CA 15-3 on EV glycan detection[45,46]. As ConA-labeled CA 125 and CA 15-3 could be efficiently filtered out with negligible signal, the spiking of CA 125 (35 U mL⁻¹) or CA 15-3 (25 U mL⁻¹) into healthy donor (HD) plasma

did not significantly increase the fluorescence intensity of HD plasma measured by EVLET (Supplementary Fig. 16). In comparison, the spiking of MDA-MB-231 EVs ($1.4 \times 10^8$ EVs per assay) led to a noticeable increase in fluorescence signal of HD plasma. EVLET was also applied to detect HD plasma sample mixed with LDL (2.7 mg mL⁻¹)[47] or chylomicrons (CM, 1.3 mg mL⁻¹)[48–50]. As shown in Supplementary Fig. 17, there was no significant difference in signal intensity between the HD sample and the spiked sample, indicating that lipoproteins have a minimal effect on EV glycan detection. Collectively, our EVLET system enabled sensitive, specific and quantitative analysis of EV glycans.

## EVLET for profiling EV glycans

We profiled the expression of glycans on EVs derived from different breast cell lines (MDA-MB-231, MDA-MB-453, BT-474, MCF-7 and MCF-10A) by EVLET using a panel of lectins (ConA, WGA, RCA I, SBA and UEA I). Figure 5a, b showed that EV glycan patterns were varied across different types of EVs, which were also confirmed by lectin-based ELISA (Pearson's correlation coefficient $r = 0.90$, $p < 0.0001$, Fig. 5c and Supplementary Fig. 18). Interestingly, EVs from aggressive breast cancer cell lines such as MDA-MB-231 (TNBC) and MDA-MB-453 (TNBC) had higher levels of ConA and WGA and lower levels of RCA I, SBA and UEA I when compared to less aggressive breast cancer cell lines such as BT-474 (Luminal B, HER2 + ) and MCF-7 (Luminal A) (Fig. 5b). Flow cytometry analysis of glycan profiles of cell lines indicated a good correlation ($r = 0.76$, $p < 0.0001$) between EVs and their parental cells (Fig. 5d and Supplementary Fig. 19). Using hierarchical clustering, different types of EVs were grouped into two main clusters: TNBC-derived EVs (MDA-MB-231 EVs and MDA-MB-453 EVs) in one cluster and luminal-like breast cancer-derived EVs (BT-474 EVs and MCF-7 EVs) in the other cluster (Fig. 5e). Dimensional reduction of all possible combinations of 3–5 lectins by principal component analysis (PCA) was implemented to determine the inter-class distance across 3 groups (TNBC EVs, other BC EVs and benign EVs). The largest inter-class distance was attained by the combination of ConA, WGA and RCA I that was selected for further experiments (Fig. 5f and Supplementary Fig. 20).

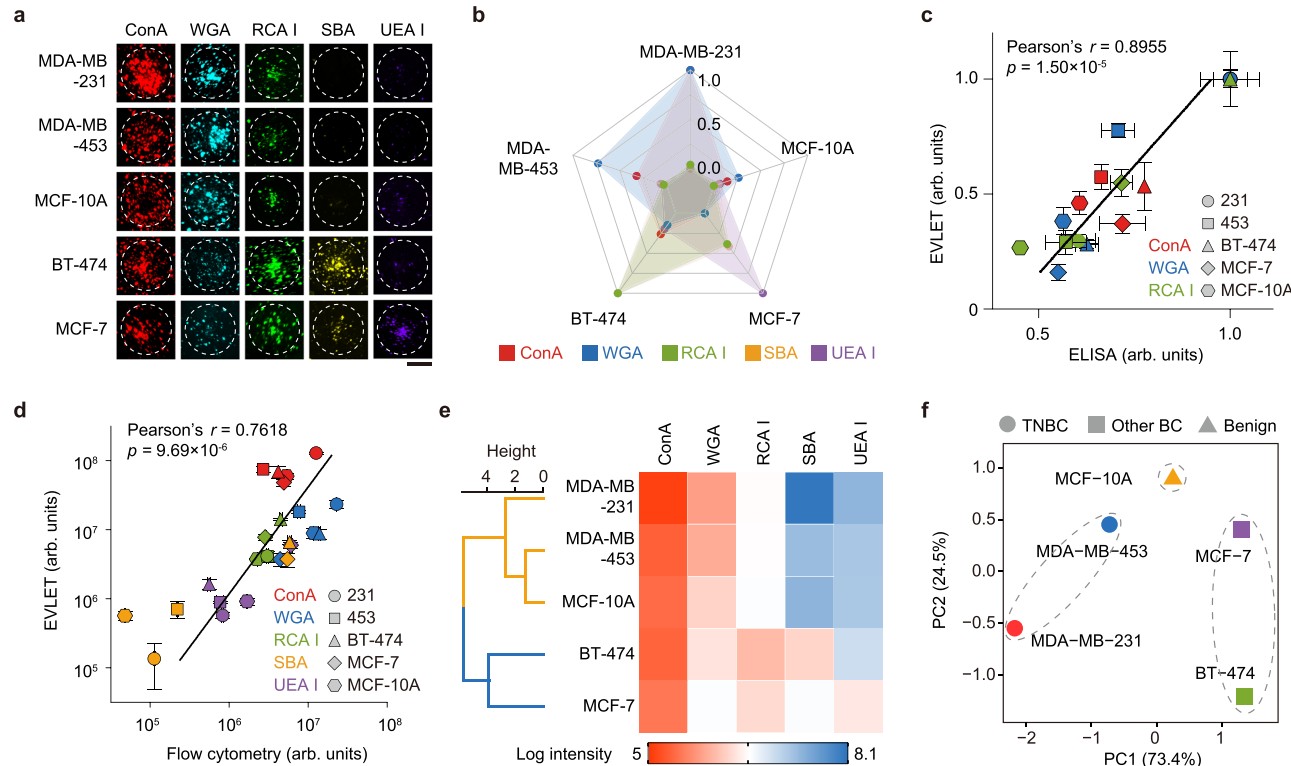

**Fig. 5 | EVLET for the measurement of EV glycan profiles using a panel of lectins.** **a** Fluorescence images of the accumulated cell line-derived EVs conjugated with different lectins. The representative images are shown from three independent repeats. Scale bar, 50 µm. **b** Radar plot showing EV glycan profiles. **c** Good correlation between the glycan pattern of EVs measured by EVLET and lectin-based ELISA ($n = 3$ samples for each cell line EVs). Pearson's $r$ and $P$-value are indicated. **d** Good correlation between the glycan pattern of EVs and that of cells measured by flow cytometry ($n = 3$ samples for each cell line EVs). Pearson's $r$ and $P$-value are indicated. **e** Hierarchical clustering of all lectins. **f** PCA showing the discrimination of 3 EV groups using the combination of ConA, WGA and RCA I. $P$-values for Pearson's correlation analyses were determined by two-sided test (**c**, **d**). Error bars represent the mean ± s.d. in (**c**, **d**). Source data are provided as a Source Data file.

## Establishing an EV glycan signature for TNBC detection

To demonstrate the application of EVLET for clinical detection of TNBC, we collected 64 plasma samples including 20 TNBC patients, 19 patients with other BC subtypes and 25 age-matched female HDs in the training cohort (Supplementary Figs. 21, 22 and Supplementary Table 4). A panel of 3 lectins (ConA, WGA and RCA I) was used to detect the EV surface glycan profiles by EVLET (Fig. 6a). Plasma samples from BC patients possessed higher levels of lectin-labeled EVs than HDs ($p < 0.0001$) (Supplementary Fig. 23). For TNBC patients, the signal of WGA-conjugated EVs was significantly higher ($p = 0.0083$) than patients with other BC subtypes, and the signal of RCA I-conjugated EVs was lower in TNBC cohort ($p < 0.0001$, Supplementary Fig. 23). Receiver operating characteristic (ROC) analysis showed that ConA, WGA and RCA I achieved an area under ROC curve (AUC) of 0.9303 (95% confidential interval (95% CI): 0.8705−0.9900), 0.9622 (95% CI: 0.9306−1.0000) and 0.9108 (95% CI: 0.8423−0.9792) for BC and HD discrimination (Supplementary Fig. 24). However, individual lectins showed moderate overall accuracy for differentiating TNBC, other BC subtypes and HD (61% for ConA, 70% for WGA and 77% for RCA I) (Supplementary Fig. 25). There were no strong correlations between any pair of lectins for detecting EV glycans (Supplementary Fig. 26). Plot of EV glycan profiles as the levels of 3 lectins using t-SNE visualization revealed the clustering of TNBC, other BC subtypes and HDs into three groups (Supplementary Fig. 27). For TNBC detection in the training cohort, an EV glycan signature (TNBC$^{EGD}$) as the weighted sum of signal intensities of 3 lectins by LDA was used for discriminating TNBC, other BC subtypes and HDs with an overall accuracy of 91% (Fig. 6b, c). In an independent validation cohort containing 32 plasma samples (8 TNBC patients, 11 other BC patients and 13 HDs), the TNBC$^{EGD}$ signature showed excellent performance (91% accuracy) for classifying TNBC, other BC and HDs (Fig. 6d and Supplementary Table 5).

## EV glycan signature for therapeutic response assessment

We further investigated whether EV glycan profiles could be employed for assessing therapeutic response in TNBC. Cisplatin, one of the most commonly used chemotherapy drugs for TNBC, was selected. Cisplatin treatment of MDA-MB-231 cells engendered a dose-dependent decrease in the cell viability, and the IC$_{50}$ value (half-maximal inhibitory concentration) of cisplatin was determined as 80 µM for MDA-MB-231 cells (Supplementary Fig. 28). Moreover, the decreased cell viability after cisplatin treatment was correlated with declined fluorescence intensities of ConA-, WGA-, or RCA I-labeled MDA-MB-231 cells measured by flow cytometry (Supplementary Figs. 29, 30). The levels of ConA-, WGA- and RCA I-conjugated MDA-MB-231 EVs secreted by cisplatin-treated cells (unified to equal EV concentration) were also decreased as detected by EVLET (Supplementary Fig. 31). We observed a strong correlation in fluorescence intensities between lectin-labeled MDA-MB-231 cells and the secreted EVs (Supplementary Fig. 31c), suggesting that EV glycan profiles could be considered as a potent indictor of drug efficacy.

For longitudinal monitoring of therapeutic response, we used the EVLET system to profile EV glycans in plasma samples of TNBC patients ($n = 13$) before (baseline) and after chemotherapy (Fig. 7a and Supplementary Tables 6, 7). The EV glycan signature for therapeutic response monitoring (TNBC$^{EGM}$ signature) was established by LDA as the weighted sum of ConA, WGA and RCA I. Among the 25 follow-up time points for 13 TNBC patients, the TNBC$^{EGM}$ signature showed better concordance with the therapeutic response (partial response, PR; stable disease, SD; progressive disease, PD) than the most widely used

Article

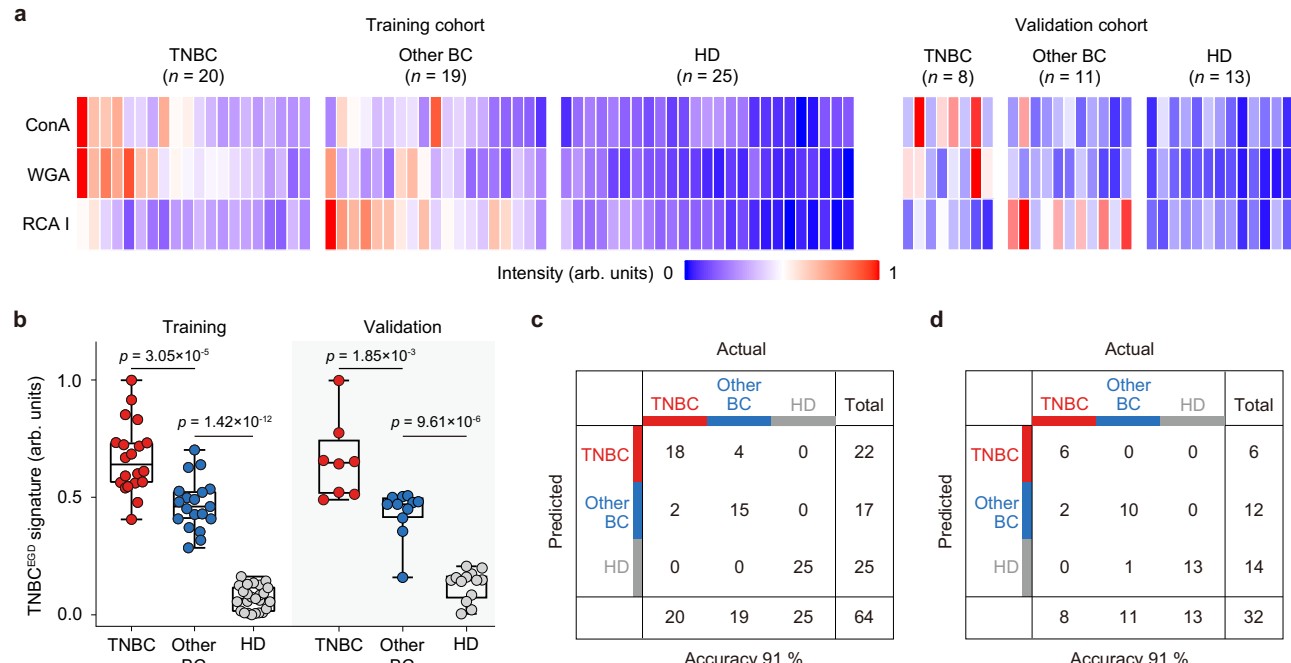

**Fig. 6 | EV glycan signature for TNBC detection. a** Heatmap of EV glycan profiles in the training cohort (20 TNBC patients, 19 other BC patients and 25 HDs) and the validation cohort (8 TNBC patients, 11 other BC patients and 13 HDs) as measured by EVLET using a panel of 3 lectins (ConA, WGA and RCA I). **b** The TNBC[EGD] signature in differentiating TNBC, other BC subtypes and HD for the training ($n = 20$ samples for TNBC, $n = 19$ samples for other BC subtypes and $n = 25$ samples for HDs) and validation cohorts ($n = 8$ samples for TNBC, $n = 11$ sample for other BC subtypes and

$n = 13$ samples for HDs). **c** Confusion matrix indicating the performance of the TNBC[EGD] signature for the training cohort. **d** Confusion matrix indicating the performance of the TNBC[EGD] signature for the validation cohort. Statistical differences were determined by two-sided, nonparametric Mann–Whitney test (**b**). *P* values are indicated in the chart. Error bars represent the mean ± s.d. in (**b**). The central line, box and error bar indicate the median, inter-quartile range (Q1 and Q3) and min-max range, respectively, in (**b**). Source data are provided as a Source Data file.

serum marker, CA 15-3. Representatively, patient P1 experienced PD at Day 150 and became PR after receiving a new line of treatment, which was consistently reflected by the increase and subsequent decrease of the TNBC[EGM] signature. In contrast, the level of serum CA 15-3 continuously decreased even at the time of PD. Notably, the TNBC[EGM] signature was significantly higher in patients at PD ($n = 10$) than those at PR/SD ($n = 15$) ($p < 0.0001$) (Fig. 7b, c), resulting in an accuracy of 96% in differentiating PD from PR/SD (Fig. 7d). In comparison, individual lectins (ConA, WGA and RCA I) and serum CA 15-3 displayed lower accuracies of 52%, 80%, 88% and 72%, for monitoring treatment response (Supplementary Figs. 32, 33). To evaluate the long-term outcome of treated TNBC patients, responders were identified as those who received at least 3 months therapy and did not encounter with PD within 6 months, and non-responders were patients who had PD within 6 months. All the non-responders ($n = 8$) showed elevated levels of TNBC[EGM] signature at the time of PD, while all the responders ($n = 5$) had decreased levels of TNBC[EGM] signature at the 6-month follow-up (Fig. 7e). The drug efficacy index ($\eta_{EV}$) defined as the temporal change in 1/(TNBC[EGM] signature) had an AUC of 1.0000 (95% CI: 1.0000 to 1.0000) for discriminating between responders and non-responders, which was better than serum CA 15-3 (Fig. 7f, g).

**EV glycan signature for prognosis of TNBC**

We also explored the potential of EV glycans for predicting progression-free survival (PFS) in TNBC patients (Fig. 8). EVLET was used to profile EV glycans in plasma samples from 25 metastatic TNBC patients before receiving salvage treatment (the baseline, Supplementary Table 8). The EV glycan signature for TNBC prognosis (TNBC[EGP] signature) was represented as the weighted sum of ConA, WGA and RCA I on EVs at the baseline. After a median follow-up period of 151 days (43–340 days), we discovered a strong association between the high level (>= median value) of TNBC[EGP] signature and poor PFS by

the Kaplan-Meier analysis, with statistical significance according to log-rank test ($p = 0.040$, Fig. 8a). TNBC patients with high expression of TNBC[EGP] signature (>= median) had a median PFS of 149 days and those with low expression (< median) had a median PFS of 205 days. Moreover, Cox proportional hazard regression using a univariate model identified the TNBC[EGP] signature as a better prognostic marker (hazard ratio (HR) = 6.17, 95% CI = 1.32−28.87, $p = 0.021$) than serum CA 15-3 (log-rank test: $p = 0.547$; HR = 1.02, 95% CI = 1.01−1.04, $p = 0.032$) (Fig. 8b, Supplementary Fig. 34 and Supplementary Table 9). Multivariate Cox proportional hazard regression further revealed that the TNBC[EGP] signature remained an independent predictor (HR = 6.16, 95% CI = 1.02−37.32, $p = 0.048$) after adjusting for serum CA 15-3, immunohistochemical status of Ki67 and age. To validate the prognostic value of glycans in TCGA-BRCA TNBC patients, we constructed an RNA signature based on mRNAs encoding glycosyltransferases or glycosidases that are related to glycans recognized by ConA, WGA and RCA. Similar to the TNBC[EGP] signature, prognostic association of RNA signature with overall survival was acquired by Kaplan-Meier analysis (log-rank test: $p = 0.029$, Fig. 8c).

## Discussion

Tumor-derived EVs in blood plasma have increasingly received attention due to their abundance, high stability and rich molecular cargos inherited from parental cells. However, most EV-based liquid biopsy studies focus on the detection and analysis of proteins or nucleic acids carried by EVs, while the potential of EV glycans is rarely investigated. Cancer-associated glycans, including α-2,6 sialic acid residues, high mannose and complex type *N*-glycans, are abundantly expressed on EVs[22,29]. EV glycan signature in cancer ascites has been used for prognosis stratification of patients with colorectal and gastric cancers[20,31]. Monitoring glycosylation alterations of EVs secreted by chemotherapy-induced senescent TNBC cells provides valuable

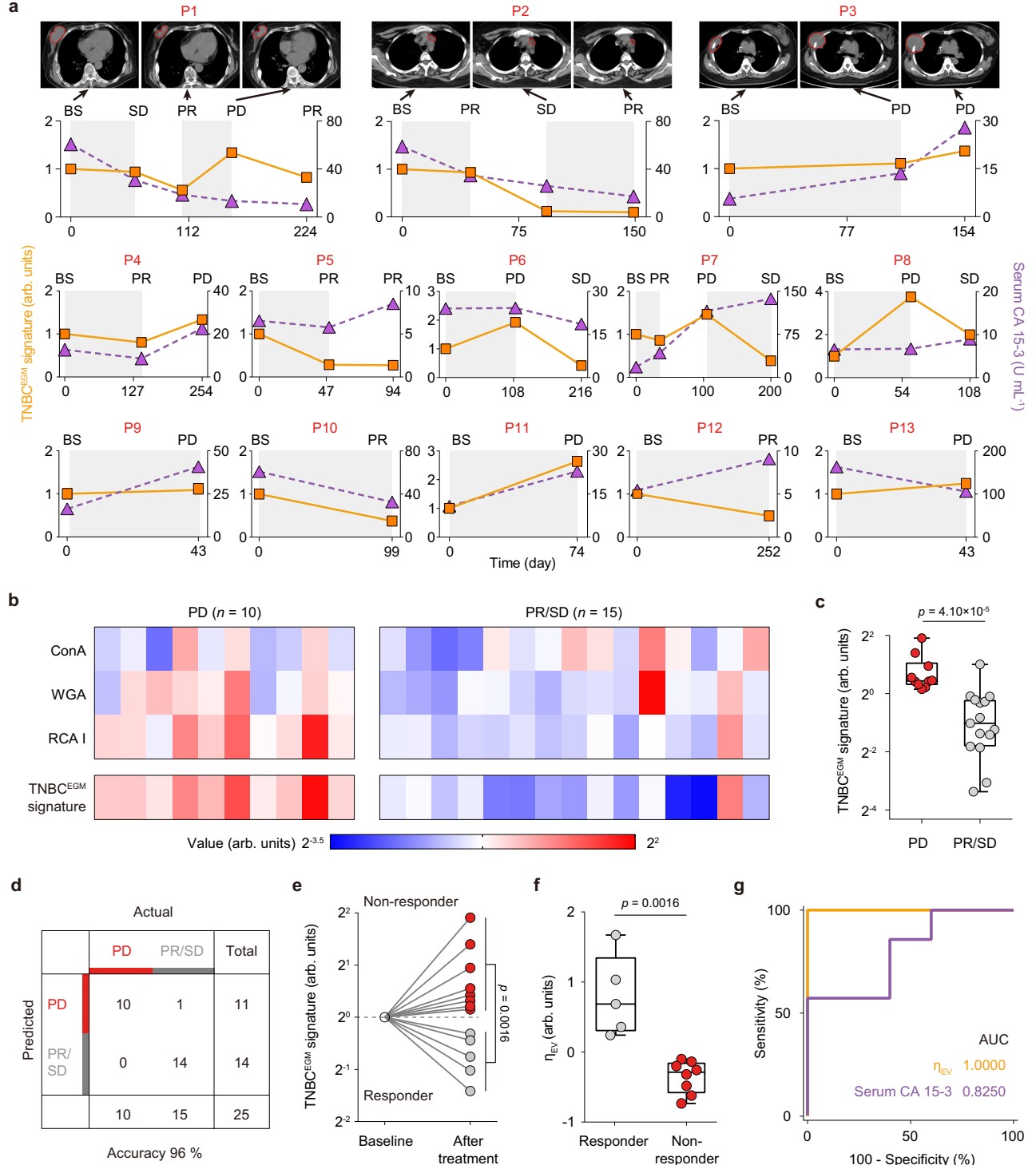

**Fig. 7 | EV glycan signature (TNBC^EGM signature) for monitoring therapeutic response. a** CT images, TNBC^EGM signature and serum CA 15-3 for longitudinal monitoring of therapeutic response in a cohort of 13 metastatic TNBC patients. The treatment response at various times was indicated. BS indicates baseline, PR indicates partial response, SD indicates stable disease and PD indicates progressive disease. **b** Heatmap of individual lectins (ConA, WGA and RCA I) and the TNBC^EGM signature between the different treatment response groups (PD versus PR/SD). **c** TNBC^EGM signature for discriminating between PD ($n = 10$) and PR/SD ($n = 15$). **d** Confusion matrix showing the accuracy of TNBC^EGM signature in differentiating

PD from PR/SD. **e** TNBC^EGM signature and (**f**), Drug efficacy index ($\eta_{EV}$) for discriminating between responders ($n = 5$) and non-responders ($n = 8$) after treatment. **g** ROC curves showing that $\eta_{EV}$ had better performance for treatment stratification than serum CA 15-3. Statistical differences were determined by two-sided, non-parametric Mann–Whitney test (**c**, **e**, **f**). *P*-values are indicated in the chart. The central line, box and error bar indicate the median, inter-quartile range (Q1 and Q3) and min-max range, respectively (**c**, **f**). Source data are provided as a Source Data file.

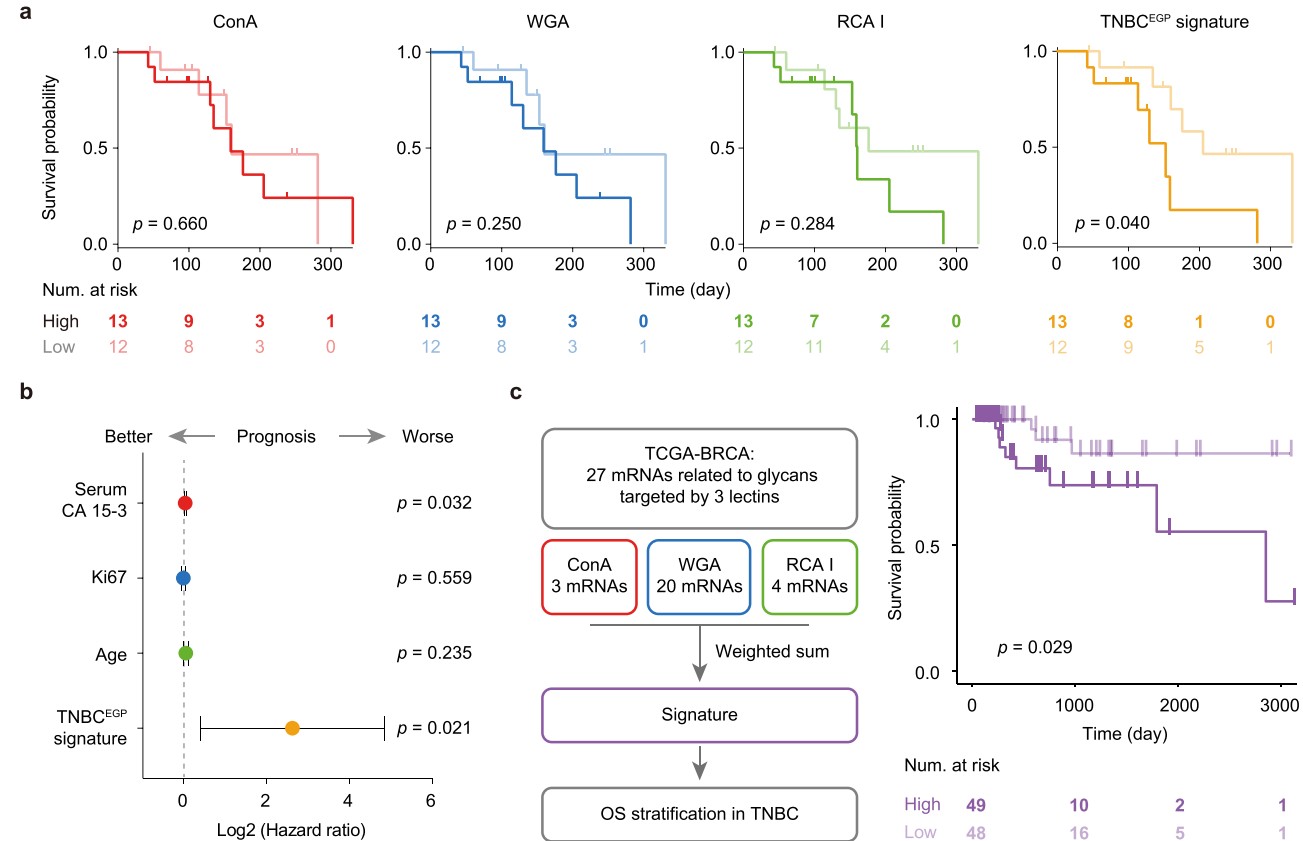

**Fig. 8 | EV glycan signature (TNBC^EGP signature) for prediction of PFS in TNBC.**
**a** Kaplan-Meier analysis showing the performance of EV glycans detected by ConA, WGA, RCA I and the performance of TNBC^EGP signature in stratifying TNBC patients ($n = 25$) with long or short PFS. The baseline level (high or low) was stratified according to the median value. **b** Forest plot showing the prognostic effect (hazard ratio, HR) of TNBC^EGP signature compared to other factors (serum CA 15-3, Ki67 and age) based on the Cox regression survival analyses for PFS of 25 TNBC patients. Two-sided test was used without adjustments for multiple comparisons.

**c** Validation of the prognostic value of EV glycan profiles for the TNBC cohort in TCGA-BRCA database. An RNA signature in analogy to the TNBC^EGP signature was constructed. The overall survival of TNBC patients in the TCGA-BRCA database was analyzed by the RNA signature. The significance of difference was calculated by a two-sided log-rank test (**a**, **c**). Centers indicate the HR and error bars indicate the 95 % confidential intervals (95 % CI). **b** P-values are indicated in the chart (**a**–**c**). Source data are provided as a Source Data file.

insight into the prediction of drug resistance and treatment efficacy[51,52]. Heavy *N*-glycosylation of PD-L1 on melanoma EVs facilitates the recognition and deactivation of PD-1^+ CD8^+ T cells, playing a vital role in inducing an immunosuppressive tumor microenvironment[23,53]. Thus, EV glycans can be considered as a new hallmark of cancer, showing great potential for non-invasive cancer detection, monitoring and prognosis. However, the existence of a large number of interfering substances such as soluble glycoproteins and various types of lipoproteins (LDL, VLDL, etc.) in clinical samples hampered the precise detection of EV glycans[54,55]. Although ultracentrifugation has been widely used for EV isolation, it suffers from co-precipitation of protein aggregates and relatively low recovery of EVs, affecting the downstream detection of EV glycans[56-58]. In addition, the requirement of expensive instrumentation and skilled operators for EV glycan analysis limits its clinical use for cancer detection.

Here, we developed an EVLET system composed of VMF and thermophoretic assay for the profiling of EV glycans directly from clinical plasma samples in a rapid, cost-effective manner. VMF obtained high-purity EVs by effectively removing >99% of lipoproteins and unbound lectins in 10 min. The sensitivity of thermophoretic assay was 2 orders of magnitude higher than lectin-based ELISA. The EVLET system enabled quantification of EV glycans in plasma of cancer patients in less than 100 min and currently cost $15 per patient sample. We showed that the EV glycan signature representing a weighted sum of 3 lectins (ConA, WGA and RCA I) could be used for clinical detection of TNBC in

a pilot cohort study. Moreover, the EV glycan signature was superior to serum CA 15-3, the most widely used serum marker in breast cancer[59], for therapeutic response monitoring and prognosis of TNBC patients.

Despite these promising findings, we believe that the EVLET system could be further improved to maximize its clinical application. First, our method detects the expression of cancer-associated glycan on EV surfaces without the knowledge of the protein or lipid to which the glycan is attached. The combination of lectins and antibodies/aptamers could be attempted for the detection of glycosylation of specific proteins or lipids. Second, the EVLET system analyzes one sample labeled with one lectin per run. For multiplexed detection of EV glycans, technical improvements including the parallelization of multiple filtration systems, the use of a number of lectins conjugated with different fluorophores and the design of multi-beam laser for high-throughput detection will be exploited. Finally, the cohort size ($n = 135$) in our study was relatively small. Future studies could expand the number of cancer patients to validate the clinical utility of EV glycan profiles for cancer diagnosis, monitoring and prognosis. We anticipate that the EVLET system can provide a significant contribution to EV glycan analysis for a wide range of clinical practice.

## Methods
### Study design
The aim of this study was to investigate EV glycans as potential biomarkers for diagnosis, response assessment and prognostics of

metastatic TNBC using EVLET. Five lectins were first selected to target glycan epitopes associated with TNBC by the data mining based on the TCGA breast cancer dataset. The glycans on EVs derived from different breast cell lines were profiled by EVLET using the five lectins, and three lectins (ConA, WGA, and RCA I) were proved to be sufficient for classifying cell line-derived EVs. In a clinical study, 135 plasma samples from TNBC patients, patients with other BC subtypes and age-matched HDs were subjected to EVLET measurement to obtain the EV glycan profiles. Machine learning was used to establish the EV glycan signature based on the signal intensities of three lectins for TNBC management. The clinical study compiled with all relevant ethical regulations and was approved by the Ethics Committee of the Fifth Medical Center of PLA General Hospital. All participants provided written informed consent.

## Reagents and materials

FITC-conjugated lectins (concanavalin A, ConA; wheat germ agglutinin, WGA; ricinus communis agglutinin 1, RCA I; soybean agglutinin, SBA; ulex europaeus agglutinin 1, UEA I) were selected for profiling EV glycans. WGA was purchased from Sigma Aldrich (USA) and the other 4 lectins were purchased from Vector Laboratories (USA). The glycan specificities of the used lectins were summarized in Supplementary Table 1. L-15 medium was purchased from Keynentec (China). Dulbecco's Modified Eagle's Medium (DMEM), RPMI-1640 medium, fetal bovine serum (FBS) and phosphate buffer saline (PBS) were purchased from Gibco (USA). MEGM BulletKit (CC-3151 & CC-4136) medium was purchased from LONZA (USA). Penicillin/streptomycin was purchased from Wisent (Canada). The filter with a nominal pore size of 0.45 μm was purchased from Millipore (USA). Anodic aluminum oxide (AAO) membrane with a nominal pore size of 20 nm was purchased from Whatman (USA). Anti-adherence rinsing solution was purchased from Stemcell (Canada). BCA Protein Assay Kit was purchased from Beyotime (China). LDL ELISA kit and VLDL ELISA kit were purchased from Cloud-Clone (China). DiD (DiIC18(5); 1,1'-dioctadecyl-3,3,3',3'- tetramethylindodicarbocyanine, 4-chlorobenzenesulfonate salt) was purchased from Thermo Fisher Scientific (USA). PNGaseF was purchased from New England Biolabs (USA). CA 125 and CA 15-3 proteins were purchased from CanAg Diagnostics (Sweden). CM was purchased from Camilo (China). LDL, standard ladder (PR1910) and Tween 20 were purchased from Solarbio (China). Gel (G2043) was purchased from Servicebio (China). Anti-CD81 antibody (ab109201), anti-Apolipoprotein B (ApoB) antibody and horseradish peroxidase (HRP)-conjugated anti-rabbit IgG (ab6721) were purchased from Abcam (UK). Anti-HSP90 antibody (207258-T32) was purchased from Sino Biological (China). Anti-calnexin antibody (bsm-52639R) was purchased from Bioss (China). Polystyrene 96-well microplate and Fixable Viability Dye eFluor™ 660 were purchased from Thermo Fisher Scientific (USA). Cisplatin was purchased from Macklin (China).

## Cell lines

Human breast cancer cell lines (MDA-MB-231, HTB-26; MDA-MB-453, HTB-131; BT-474, HTB-20; and MCF-7, HTB-22) and human benign breast epithelial cell line MCF-10A (CRL-10317) were obtained from American Type Culture Collection (ATCC, USA). MDA-MB-231 and MDA-MB-453 cells were cultured in L-15 medium at 37 °C. BT-474 cells were cultured in DMEM at 37 °C with 5% $CO_2$. MCF-7 cells were cultured in RPMI-1640 medium at 37 °C with 5% $CO_2$. MCF-10A cells were cultured in MEGM BulletKit (CC-3151 & CC-4136) medium at 37 °C with 5% $CO_2$. All media except for MEGM BulletKit medium were supplemented with 10% FBS and 1% penicillin/streptomycin.

## EV isolation by ultracentrifugation

For cell line EV isolation, cells were cultured to reach a confluency of 70% prior to EV isolation. The collected media (240 mL) were centrifuged at 2000 × g for 10 min by twice to remove cells and large debris. The resulting supernatant was filtered through a 0.45 μm Millipore filter

and ultracentrifuged at 100,000 × g for 3 h. The EVs were obtained by suspending the pellet in 1600 μL of 1× PBS. For plasma EV isolation, the plasma samples (2 mL) were diluted to 27 mL using 1× PBS and centrifuged at 2000 × g for 10 min by twice. The resulting supernatant was filtered through a 0.45 μm Millipore filter and ultracentrifuged at 100,000 × g for 3 h. The EVs were obtained by suspending the pellet in 200 μL of 1× PBS. Before ultracentrifugation, the weight difference between every pair of tubes was set to be smaller than 0.05 g. The collected cell line EVs and plasma EVs were stored at −80 °C before use.

## Size exclusion chromatography (SEC)

SEC was used to process the plasma samples. SEC column (qEV2, 35 nm) was flushed by 45 mL PBS before use. 2 mL plasma was directly loaded into the column. The loaded sample was eluted with PBS, and EV-containing fractions (14–22 mL) were pooled. For reuse, the column post plasma processing was sequentially washed with 90 mL of PBS, 2 mL of 0.5 M NaOH and 90 mL of PBS to remove the residual proteins.

## Design and implementation of VMF

A customized VMF system was shown in Supplementary Figs. 1, 2. The core component was a funnel-shaped filter that was fabricated by 3D printing. The cone-shaped cavity of the filter served as the reservoir of sample fluid, and an AAO membrane (20 nm in nominal pore size) was mounted at the bottom of the cavity. To improve the filtration performance, a piezoelectric oscillator was integrated concentrically under the AAO membrane to generate high-frequency (3–9 kHz) membrane vibration, leading to the levitation of EVs to reduce the membrane fouling and the damage of EVs. The bottom neck of the filter was connected with a syringe pump at a negative pressure mode ( ~ 50 kPa). To automate the entire filtration workflow including filtration, washing and retrieval of EVs, a customized system was constructed by integrating the AAO membrane filter, the high-frequency piezoelectric oscillator, the square-wave generator, the negative pressure-controlled pneumatic driving unit, the pipetting unit, the reservoirs for washing buffer and waste and the controller.

The AAO membranes were blocked by anti-adherence rinsing solution overnight, washed by DI water, and dried by air prior to filtration. To process cell line EVs, 200 μL of EVs ($10^7$ μL$^{-1}$) was incubated with FITC-conjugated lectin for 1 h at room temperature and then loaded into the AAO membrane for vibrating filtration. The filtration procedure was as follow: (i) the sample was filtrated through the AAO membrane; (ii) the filtered sample was washed using 150 μL 1× PBS by twice; (iii) the isolated EVs were finally resuspended into 100 μL 1× PBS by automatic pipetting (1 Hz for 3 min). During the entire procedure (filtration, washing and retrieval), an oscillator (Shenglian, China) mounted under the AAO membrane was supplied with a high-frequency (6 kHz) alternating current (25 V) to generate vibration. For the filtration and washing process, a negative pressure of 50 kPa was added to drive the sample fluid pass through the membrane. To optimize the vibration frequency, MDA-MB-231 EVs were processed by the filtration system with different frequencies (0, 3, 6, and 9 kHz), and the other procedure was same as above.

To process plasma samples, 2 μL of plasma was diluted in 1.5 mL 1× PBS before filtration, and the other procedure was same as above.

## Scanning electron microscopy (SEM)

To characterize the real pore size, the blank AAO membrane was coated with a 3 nm Pt layer to increase the contrast and observed on an SEM (S4800, Hitachi, Japan). The size of > 90% of nanopores was 20–49 nm measured by SEM.

To characterize the isolated EVs on AAO membrane, MDA-MB-231 EVs (200 μL, $10^7$ μL$^{-1}$) or plasma samples (2 μL, diluted in 1.5 mL 1× PBS) were processed by VMF and washed using 150 μL of 5% glucose solution by twice, and the other procedure was same as the established protocol. The AAO membrane was frozen at −80 °C, and then freeze

dried. The AAO membrane was coated with a 3 nm Pt layer to increase the contrast and observed by SEM.

## Transmission electron microscopy (TEM)
MDA-MB-231 EVs ($10^8$ $\mu L^{-1}$) before VMF were adsorbed on Formvar/carbon-coated copper grids for 20 min. MDA-MB-231 EVs after VMF (~$8 \times 10^6$ $\mu L^{-1}$) were adsorbed on Formvar/carbon-coated copper grids for 1.5 h. The adsorbed EVs were then stained by 5 drops of 3% uranyl acetate. After drying, the samples were observed on a Tecnai G2 20 S-TWIN TEM (FEI, USA) at 200 kV.

## Nanoparticle tracking analysis (NTA)
To characterize the size distribution and concentration, samples containing EVs were detected using NTA (NanoSight NS300, Malvern Instrument, England) at $23 \pm 2$ °C. Particle concentrations were adjusted to $2 \times 10^5 - 8 \times 10^5$ particles $\mu L^{-1}$ to circumvent the artifacts arising from the overlap of particle trajectories. A high camera level (15, maximum = 16) and a low threshold value (5, maximum = 50) was set to avoid underestimation in particle concentration. The data of size distribution were captured and analyzed with the NTA 3.4 Analytical Software Suite. To evaluate the performance of NTA, polystyrene (PS) particles (Invitrogen, USA) of three different nominal sizes, 41 nm, 47 nm and 100 nm, were subjected to NTA measurement. The sizes of these particles were accurately identified (Supplementary Fig. 7a). Additionally, the accuracy of concentration quantification was verified by detecting 47 nm particles with varying concentrations ($2 \times 10^5$ $\mu L^{-1}$–$8 \times 10^5$ $\mu L^{-1}$). The concentrations measured by NTA were aligned well with the actual concentrations ($R^2 = 0.9160$) (Supplementary Fig. 7b).

## Removal of plasma proteins and lipoproteins by different methods
To evaluate the removal efficiency of proteins, plasma samples were first processed by VMF (2 μL plasma per sample), UC (2 mL plasma per sample), or SEC (2 mL plasma per sample) following the established protocol. The recovered samples and unprocessed plasma samples were analyzed using BCA Protein Assay Kit. Briefly, 20 μL of samples were pipetted into microplate wells and mixed with 200 μL of BCA working reagent. After incubation for 30 min at 37 °C, the optical density (OD) of the solution at 562 nm was immediately detected using a microplate reader (Synergy H1, Biotek, USA). The protein concentrations of the recovered samples and unprocessed plasma samples were calculated based on the standard curve. The EV purity was determined as the number of particles per μg protein.

To evaluate the removal efficiency of lipoproteins (LDL and VLDL), plasma samples were first processed by VMF (2 μL plasma per sample), UC (2 mL plasma per sample), or SEC (2 mL plasma per sample) following the established protocol. The recovered samples and unprocessed plasma samples were analyzed using LDL or VLDL ELISA kits. Briefly, the samples were adsorbed onto ELISA plates coated with anti-LDL or anti-VLDL antibody for 1 h at 37 °C (100 μL for each well). The supernatant was discarded, and the plates were incubated with 100 μL of biotin-conjugated LDL or VLDL antibody for 1 h at 37 °C. After washing with washing buffer for 3 times, 100 μL of SA-conjugated HRP was added and incubated for 30 min at 37 °C. The supernatant was discarded again, and ELISA plates were washed with washing buffer for 5 times. 90 μL of TMB substrate solution was added and incubated for 15 min at 37 °C. The reaction was terminated by the addition of 50 μL of stop solution and the OD of the solution at 450 nm was immediately detected using a microplate reader. The lipoprotein concentrations of the recovered samples and unprocessed plasma samples were calculated based on the standard curve.

## Western blot (WB)
The plasma samples were processed by VMF (2 μL plasma for one run of filtration, and the recovered samples were concentrated to meet the

protein loading amount for WB analysis), ultracentrifugation (2 mL plasma per sample), or SEC (2 mL plasma per sample) following the established protocol. The recovered samples were lysed by 1 × SDS-PAGE loading buffer at 100 °C for 10 min. Samples and standard ladder were loaded onto a gel with equal protein amounts (1.5 μg), separated by electrophoresis (80 V for 30 min and 140 V for 60 min), and transferred onto a PVDF membrane. The PVDF membrane was blocked with TBST buffer (tris-buffered saline and 0.1% Tween 20) containing 5% BSA for 30 min at room temperature. The blocked PVDF membrane was incubated with anti-CD81 antibody (1/1000), anti-HSP90 antibody (1/500), anti-Calnexin antibody (1/500) and anti-ApoB antibody (1/10000) at 4 °C overnight, followed by washing three times for 10 min using TBST buffer. The PVDF membrane was then incubated with HRP-conjugated anti-rabbit IgG (1/5000) at room temperature for 1 h, followed by washing three times for 10 min using TBST buffer. The protein bands were visualized using chemiluminescence reagent and observed on a Tanon 5200 multi-gel analysis (Tanon Shanghai, China).

## Quantification of the removal of free lectin
200 μL of FITC-conjugated ConA solution (5 μg ml$^{-1}$) was processed by VMF. The fluorescence of the filtrate was quantified by a fluorescence spectrometer (FluoroMax +, HORIBA, USA). The remained concentration of ConA in the filtrate was calculated based on the established standard curve (0.008–25 μg ml$^{-1}$).

## Fluorescence colocalization analysis of EVs
To determine the percentage of lectin-conjugated EVs derived from different cell lines, FITC-ConA (5 μg mL$^{-1}$) was used to label MDA-MB-231 EVs and MCF-10A EVs ($10^7$ $\mu L^{-1}$, 200 μL) at room temperature for 1 h. The EV samples were then filtered through VMF (6 kHz, 25 V, 50 kPa), washed twice with 150 μL PBS, and retrieved in 100 μL PBS. The recovered samples were treated with 4 nM lipophilic DiD to label all EVs at 37 °C for 20 min. The fluorescence colocalization of DiD and FITC signals was analyzed under a fluorescence microscope (DMi8, Leica, Germany) with a 100 × objective, and the images were captured with an sCMOS camera (95B, Photometrics, Canada) using 2 × 2-pixel binning and a 50 ms exposure time. For MDA-MB-231 EVs, the percent of ConA-labeled EVs was 25.8% based on 90 DiD-FITC colocalization events out of 349 EVs (DiD). For MCF-10A EVs, the percent of ConA-labeled EVs was 6.6% based on 22 DiD-FITC colocalization events from 332 EVs (DiD).

## Numerical simulation
The flow field, heat transfer and EV transportation in the microchamber were obtained by solving their government equations using a finite element method, as follows:

$$
\begin{aligned}
\nabla \cdot (\rho \mathbf{u}) &= 0 \\
\frac{\partial(\rho \mathbf{u})}{\partial t} + \nabla \cdot (\rho \mathbf{u}\mathbf{u}) &= -\nabla p + \nabla \cdot (\eta \nabla \mathbf{u}) - \rho g \alpha \Delta T \\
\frac{\partial(\rho T)}{\partial t} + \nabla \cdot (\rho \mathbf{u} T) &= \nabla \cdot (\frac{\lambda}{c_{\mathrm{p}}} \nabla T) + S \\
\frac{\partial c}{\partial t} + \nabla \cdot (\mathbf{u}c) &= \nabla \cdot (-D \nabla c - S_T D c \nabla T)
\end{aligned}
\tag{1}
$$

where $\mathbf{u}$ is the flow velocity vector, $\rho$ is the fluid density, $p$ is the pressure, $\eta$ is the dynamic viscosity, $c$ is the EV concentration, $S_T = 0.03(a/100 \text{ nm})^2$ $K^{-1}$ is the Soret coefficient of EVs with a diameter of $a$, and $D = kT/3\eta\pi a$ is the diffusion coefficient of EVs. All governing equations were solved by a finite element solver (Comsol, Femlab). Gravity force was applied to the fluid motion to enable the thermal convection induced by the thermal expansion of water. The expansion rate $\alpha$ was derived from the temperature dependence of $\rho$. The thermal conductivity $\lambda$ of water, glass and sapphire were 0.6, 1.3 and 35 W m$^{-1}$ K$^{-1}$, respectively. The heat capacity $c_{\mathrm{p}}$ of water, glass and sapphire were 4200, 761 and 755 J Kg$^{-1}$ K$^{-1}$, respectively. For the 1480 nm laser, the power was 150 mW and the diameter of focused spot was 160 μm.

The effect of size distribution of EVs was considered in the simulation. The percentages of EVs with different size ranges at a step of 20 nm were obtained from the NTA measurement of MDA-MB-231 EVs post VMF. The simulation was performed for each center size (50 nm, 70 nm, 90 nm, etc) and the enrichment ratio of total EVs was determined as the weighted sum of enrichment ratio of EVs with each center size (weights were their percentages). The simulation result indicated that EVs can be accumulated by over 2500 folds at the center of microchamber bottom within 10 min of laser irradiation.

### Thermophoretic analysis of EVLET

For thermophoretic analysis of EV glycans, the recovered lectin-labeled EVs were introduced into a microchamber with a thickness of 400 μm and a diameter of ~7 mm (fabricated by sandwiching a 400 μm-thick spacer between a 1-mm-thick glass top layer and a 1 mm-thick sapphire bottom layer). The microchamber containing 14 μL of solution was then mounted on an inverted fluorescence microscope (DMi8, Leica, Germany) with a 40× objective. The samples at the microchamber center were locally heated by a 1480 nm laser (Changchun Laser Optoelectronics Technology, China) with a power of 150 mW for 10 min for thermophoretic enrichment of EVs. The fluorescence intensities before and after laser irradiation were captured by a sCMOS (95B, Photometrics, Canada) with $2 \times 2$ pixel binning and an exposure time of 50 ms, and recorded using ImageJ (1.52a, NIH). The experiments were performed in dark to avoid photobleaching.

The specificity of EVLET for EV glycan analysis was verified with PNGase F treatment. MDA-MB-231 EVs were treated with PNGase F (10-fold dilution) in GlycoBuffer (1×) for 12 h. The deglycosylated EVs were incubated with 5 μg mL$^{-1}$ of ConA for 1 h at room temperature and then detected following the established EVLET protocol.

To evaluate the sensitivity and linear range of EVLET for detecting EV glycans, MDA-MB-231 EVs with different concentrations (0, $8 \times 10^4$, $4 \times 10^5$, $2 \times 10^6$ and $10^7$ μL$^{-1}$) were incubated with 5 μg mL$^{-1}$ of ConA for 1 h at room temperature, and then processed by the EVLET system. To confirm that thermophoretic enrichment provides an improvement in detection sensitivity, 100 μL of the ConA-labeled MDA-MB-231 EVs (0, $2 \times 10^6$, $1 \times 10^7$, $5 \times 10^7$ μL$^{-1}$) post VMF were directly detected using a microplate reader.

For EV glycan profiling, EVs derived by MDA-MB-231, MDA-MB-453, BT-474, MCF-7 and MCF-10A cells were incubated with ConA (5 μg mL$^{-1}$), WGA (20 μg mL$^{-1}$), RCA I (5 μg mL$^{-1}$), SBA (5 μg mL$^{-1}$), or UEA I (5 μg mL$^{-1}$) at room temperature for 1 h, and then detected by the EVLET system.

### Lectin-based ELISA

EVs ($10^6$ μL$^{-1}$, 100 μL) were immobilized onto 96-well ELISA plates with polystyrene substrate (437112, Thermo Scientific, USA) through passive adsorption for 2 h at room temperature[60,61]. After EV immobilization, the supernatant was discarded, and the wells were blocked using 100 μL of 1× PBS containing 1% BSA for 1 h at room temperature. After discarding the supernatant, the wells were rinsed 3 times with 100 μL of 1× PBS containing 1% BSA. Subsequently, 100 μL of solution containing lectin (5 μg mL$^{-1}$ FITC-ConA, 20 μg mL$^{-1}$ FITC-WGA, or 5 μg mL$^{-1}$ FITC-RCA I) and 1% BSA was added, and the plate was incubated for 1 h at room temperature. After labeling, the supernatant was removed, and the wells were washed 5 times with 100 μL 1× PBS containing 1% BSA. Fluorescence detection was then performed using a microplate reader (Synergy H1, Biotek, USA) with an excitation at 490 nm and an emission at 525 nm.

To evaluate the sensitivity and linear range of lectin-based ELISA for detecting EV glycans, 100 μL of MDA-MB-231 EVs with different concentrations (0, $3.13 \times 10^5$, $6.25 \times 10^5$, $1.25 \times 10^6$ and $2.5 \times 10^6$ μL$^{-1}$) were loaded onto ELISA plates, and then detected by the above protocol.

### Flow cytometry-based glycan profiling

Equal volume (200 μL, $2 \times 10^6$ cells mL$^{-1}$) of MDA-MB-231, MDA-MB-453, BT-474, MCF-7, or MCF-10A cells were labeled with a panel of 5 FITC-conjugated lectins (5 μg ml$^{-1}$ ConA, RCA I, SBA, UEA I; 20 μg ml$^{-1}$ WGA) for 1 h at 4 °C. The labeled cells were washed by 1× PBS (centrifugation at 500×g for 5 min) twice and suspended in 800 μL 1× PBS containing 2% FBS prior to the flow cytometric characterization (ACEA NovoCyte, Agilent Technologies, USA). For each sample, > 3000 cells were analyzed.

### Drug response of TNBC cells

MDA-MB-231 cells were seeded in a 96-well plate (10,000 cells per well) overnight and treated with different concentrations (0.98 μM – 1 mM) of cisplatin for 72 h. The supernatant was replaced by 100 μL mixture of CCK-8 reagent and L-15 medium (1:9, v/v) and incubated for another 2 h at 37 °C. Then OD at 450 nm was measured using a microplate reader (Synergy H1, Biotek, USA). The dose-response curve between cisplatin concentration and cell viability of MDA-MB-231 cells was drawn by Graphpad Prism. To assess the treatment response, MDA-MB-231 cells were treated with different concentrations of cisplatin corresponding to 100%, 75%, 50% and 25% cell viability for 72 h. The cell culture medium was then collected and ultracentrifuged to isolate EVs.

For flow cytometry analysis, the cells were labeled with FITC-conjugated lectins (5 μg ml$^{-1}$ ConA, RCA I; 20 μg ml$^{-1}$ WGA) for 1 h at 4 °C, followed by labelling with live/dead dye (Fixable Viability Dye eFluor™ 660, 1/1000) for 30 min at 4 °C. The labeled cells were washed by 1× PBS (centrifugation at $500 \times g$ for 5 min) twice and suspended in 800 μL 1× PBS containing 2% FBS prior to the flow cytometric characterization (ACEA NovoCyte, Agilent Technologies, USA). For each sample, > 3000 cells were analyzed. For EVLET, the EVs secreted by cisplatin-treated cells were prepared at equal concentration ($5 \times 10^6$ μL$^{-1}$, 200 μL) and labeled with FITC-conjugated lectins (5 μg ml$^{-1}$ ConA, RCA I; 20 μg ml$^{-1}$ WGA) for 1 h at room temperature. Then the samples were detected following the established EVLET protocol.

### Clinical cohort

From 2019 to 2023, 135 plasma samples from 42 TNBC patients, 30 non-TNBC patients (HR+ or HER2 + BC patients) and 38 age-matched HDs were collected from the Fifth Medical Centre of PLA General Hospital. The diagnosis cohort consisted of a training cohort ($n = 64$) and an independent validation cohort ($n = 32$). The information of the clinical samples was listed in Supplementary Tables. The patients received chemotherapy treatment in accordance with Standard of Care (the Chinese Society of Clinical Oncology Breast Cancer Guidelines). The types of therapeutic drugs mainly include platinum, taxane, navelbine, capecitabine and gemcitabine, and the treatment duration ranges from 2 to 12 cycles. None of the patients were enrolled in any clinical trial. Non-responders were defined as the patients encountered with PD within 6 months, responders were defined as the patients experienced at least 3 months therapy and did not encounter with PD within 6 months. The first appearance of PD or the last censoring event for each metastatic TNBC patient during the study period was recorded for the prediction of progression-free survival (PFS). Therapeutic response (partial response, PR; stable disease, SD; progressive disease, PD) was determined by analyzing computed tomography (CT) or magnetic resonance imaging (MRI) scans using Response Evaluation Criteria in Solid Tumors (RECIST, version 1.1)[62].

### Plasma sample collection

The clinical plasma samples were collected and processed following the Early Detection Research Network (EDRN) standard operating procedure (SOP): (i) blood samples were collected from patients into lithium-heparin plasma separator tubes (green top) and gently inverted the tube 5–8 times, (ii) the tubes were stored upright at 4 °C for

30 min, (iii) the tubes were centrifuged at 2000 × g for 30 min, (iv) the plasma was pipetted into labeled centrifuge tubes and stored at −80 °C before use.

### EVLET for clinical plasma sample

To evaluate the effects of soluble glycoproteins on EVLET assay, 2 μL of HD plasma spiked with glycoprotein (CA 125, 35 U ml$^{-1}$ or CA 15-3, 25 U ml$^{-1}$) or spiked with MDA-MB-231 EVs ($5 \times 10^8 \mu L^{-1}$) and glycoprotein (CA 125, 35 U ml$^{-1}$ or CA 15-3, 25 U mL$^{-1}$) was diluted for 10 folds and incubated with FITC-conjugated lectins (5 μg ml$^{-1}$ ConA) for 1 h at room temperature. Then 20 μL of these incubated samples were diluted in 1.5 mL 1×PBS and detected following the established EVLET protocol (14 μL of the 100 μL recovered sample was loaded into microchamber for the thermophoretic analysis).

To evaluate the effects of lipoproteins on EVLET assay, 2 μL of HD plasma was spiked with LDL (2.7 mg ml$^{-1}$) or CM (1.3 mg ml$^{-1}$) and 2 μL of EV-containing ($5 \times 10^8 \mu L^{-1}$) HD plasma was spiked with LDL (2.7 mg ml$^{-1}$) or CM (1.3 mg ml$^{-1}$). These samples were diluted for 10 folds and incubated with FITC-conjugated lectins (5 μg ml$^{-1}$ ConA) for 1 h at room temperature. These incubated samples were diluted in 1.5 mL 1×PBS and detected by EVLET.

To perform EVLET for clinical plasma samples, 20 μL of 10-fold diluted plasma was incubated with FITC-conjugated lectins (5 μg ml$^{-1}$ ConA, RCA I; 20 μg ml$^{-1}$ WGA) for 1 h at room temperature. Then these incubated samples were diluted in 1.5 mL 1×PBS and detected by EVLET.

### Establishment of EV glycan signature

For TNBC diagnosis, the TNBC$^{EGD}$ signature was defined as a weighted sum of the signal intensities of a panel of 3 lectins (ConA, WGA and RCA I) by a two-step linear discriminant analysis (LDA). The first LDA was used to discriminate BC patients and HDs and the second LDA was to classify the individuals predicted as BC into TNBC or other BC subtypes. The LDA model for diagnosis was trained using a training cohort and further tested in an independent cohort. For treatment response assessment, the TNBC$^{EGM}$ signature was defined as a weighted sum of the signal intensities of the 3 lectins by LDA. For PFS prediction, the TNBC$^{EGP}$ signature was defined as a weighted sum of the baseline levels of signal intensities of 3 lectins by LDA. LDA was performed using MASS package in R software (version 4.2.3).

### Statistical analysis

For TNBC diagnosis and prognosis, the signal intensities were linearly scaled to the range of 0 to 1 for each type of lectin. For treatment response assessment, the signal intensities were normalized by the baseline (before treatment) value for each type of lectin. For the segregation of EVs derived from different cell lines, hierarchical clustering was performed to classify groups based on the similarity in the glycan profiles detected by EVLET using 5 lectins (ConA, WGA, RCA I, SBA and UEA I). The similarity was calculated as Euclidean distance between high-dimensional points representing the signal intensities of the 5 lectins. Clustering was performed in a bottom-up manner and the samples were finally merged into two clusters. To classify cell line EVs into 3 groups (TNBC EVs, other BC EVs and benign EVs), principal component analysis (PCA) was used to perform dimensional reduction (to 2-dimensional space) of all possible combinations of 2–5 lectins. The segregation criterion was defined as that the intra-class distance (distance between samples in the same class) was smaller than the inter-class distances (distance between class centers) after dimension reduction. At least 3 lectins were required for the 3-group classification. The significance of difference in signal intensities during EVLET characterization was tested by a two-sided $t$-test. The significance of difference between signal intensities of individual lectins, EV glycan signature and serum CA 15-3 of two different clinical sample groups was tested using a two-sided, nonparametric Mann-Whitney $U$-test.

Receiver operating characteristic (ROC) curves were constructed to evaluate the accuracy for any two-class discrimination for the diagnosis and treatment monitoring of TNBC. Accuracy was defined as the overall probability that an individual was correctly classified. The 95% CIs were calculated using a binomial distribution. The prognostic values of different variables of interest, EV glycan signature and individual lectins were analyzed using Kaplan−Meier method, log-rank test, univariate Cox proportional-hazard regression and multivariate Cox proportional-hazard regression (adjusted for age, Ki67 status and serum CA 15-3). Kaplan−Meier analysis, log-rank test and Cox proportional-hazard regression were performed using R software (version 4.2.3).

### Reporting summary

Further information on research design is available in the Nature Portfolio Reporting Summary linked to this article.

## Data availability

All data generated in this study are provided in the paper, Supplementary Information and Source Data file. The expression data of mRNA transcripts in the TCGA-BRCA cohort used in this study are available in the Cancer Genome Atlas (TCGA) database under accession code phs000178 (https://portal.gdc.cancer.gov). Source data are provided with this paper.

## Code availability

We used publicly available software for the analyses. All software used in this study is described in the Methods section and the accompanying Reporting Summary.

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

## Acknowledgements

This work was supported financially by National Key R&D Program of China (No. 2020YFA0210800, 2021YFA0909400, J.S.; No. 2022YFC2403203-3, Y.W.), the National Natural Science Foundation of China (No. 22025402, 22227805, J.S.; No. T2222008, 22174030, C.L.; No. 22104026, F.T.; No. 22304005, Y.L.; No. 51972003, 52271127, Y.W.), Chinese Academy of Sciences (No. ZDBS-LY-SLH025, XDB36000000, YSBR-036, J.S.), Youth Innovation Promotion Association CAS (No. 2021036, C.L.), the China Postdoctoral Science Foundation (No. 2022M720286, Y.L.), Postdoctoral Fellowship Program of CPSF (No. GZB20230038, Y.L.) and Zhejiang Provincial Natural Science Foundation (No. LY21C050001, T.F.).

## Author contributions

J.S., L.Z., Y.W. and C.L. designed and supervised the study; Y.L., F.T. and L.Z. performed the experiments; J.S., C.L., Y.L. and J.D. analyzed and interpreted the data; S.Z., L.Q., L.B. and T.F. provided critical materials and protocols; J.S., C.L., Y.L., J.D. and Q.F. wrote the manuscript.

## Competing interests

The authors declare no competing interests.
