## [Peer Review File · Nature Communications]

Reviewers' comments:

Reviewer #1 (Remarks to the Author):

This study applies a lectin microarray to classify EVs in the context of TNBC diagnostics. The authors have adapted a conventional microarray to utilize thermophoretic capabilities. Overall the idea is pretty compelling, and even tested in >100 clinical samples. I have a few clarifications and suggestions related to perceived weaknesses of EV enrichment and characterization:

1. Please comment on the pore size of the membrane filtration overlapping with EV size. The mode of EV diameter is in the range of 50-60 nm, with some/many as small as 30-40nm. These are not perfectly rigid spheres and so some small EVs could be lost through the filter. Also LDL (20-30nm) and VLDL (30-90 nm) are heavily overlapping right in the size range of the EVs and larger than the filter pore sizes. About ~20% of the pores are larger than 40 nm, presumably a "soft" 60-70 nm EV could be compressed through here?. How could EVs be retained while the LDL/VLDL goes through? Obviously some larger EVs are retained as shown in the excellent SEM images, but then you are only examining the larger ones (>100 nm). This could be fine if the lectins on these larger EVs are the ones that carry the diagnostic information but at the least this point should be carefully clarified in the manuscript. What about chylomicrons - other LPPs that are highly abundant and glycosylated? These are seemingly ignored. How exactly are the recovery rates calculated? Just by the free FITC-ConA? How do you know it's free and not small-EV-bound? Also this wouldn't tell you how much non-ConA labeled EVs got through. What is the distribution of lectins across all EVs? Hard to believe this is a universal label.

2. The characterization of the EVs retained by the custom vibrating membrane filtration system is very weak. Only SEM (with a few EVs) and NTA (known to be very inaccurate to size and assess concentration of EVs) is performed. NTA can only see the >80 nm particles which represent less than 30% of the total number at most. The authors should refer the MISEV-2018 document which provides a comprehensive list of suggested assays that minimally is needed for accurate reporting that EVs are recovered in high yield. This is especially critical for a "new" type of EV isolation reported here. Don't get me wrong - it's a very exciting technology with great application, just need some more details on the characterization of the EVs emerging from this unit. I strongly suggest performing more accurate EV sizing in the relevant <100 nm range like resistive pulse sensing, cryo-electron microscopy (not on filter but after collection from the filter), single particle interferometric imaging, or super resolution microscopy. These other methods would be critical to support the claims.

Reviewer #2 (Remarks to the Author):

In this study, Li et al. sought to design a lectin based (EVLET) platform for diagnosis and monitoring of TNBC patients. In general, the manuscript is well written and presented, and the inclusion of longitudinal sampling is one of the strongest points in this manuscript. The concept of thermophoretic enrichment of EVs is interesting and innovative, but lacks novelty as this group has previously published this (PMID: 32134270). Furthermore, the field has moved a long way from when ultracentrifugation was the gold standard. It is now commonplace to use either/both tangential flow filtration, and size exclusion chromatography. The authors do not take this into account throughout the manuscript. As a result, there is no comparison to commonly used techniques, and is therefore not possible to draw conclusions on the performance of the purification strategy. As a result of this, as well as some more specific comments below, I do not recommend this for publication.

Specific comments:

Several sentences throughout the manuscript make claims without providing evidence/relevant citation. For example; Line 55 “Despite its feasibility, lectin microarray has not yet reached sufficient sensitivity due to the diffusion-limited mass transport of EVs to the substrate.” If this has been previously shown, please cite the relevant literature.

Line 59 “These microfluidic methods involve the fabrication of complex microfluidic devices or functional nanoparticles, thus hampering their clinical applications.” How does this hamper clinical applications?

In terms of the bioinformatic approach to identify differentially expressed proteins involved in glycosylation the authors identify 5525 mRNAs found to be differentially expressed using the TCGA database. The authors then provide evidence that simply the expression of glycosyltransferases or glycosidases can differentiate TNBC. Does this classify TNBC more accurately compared to a model incorporating the 5525 differentially expressed mRNAs? If not, this seems to be redundant as aberrant glycosylation in cancer is a well-established paradigm.

The authors are using a vibrating membrane filtration system to purify EVs with a 20 nm nominal pore size. Given the size of LDL > 20 nm and VLDL >50 nm, how do the authors claim the removal of LDL and VLDL is at > 99.1 %, and > 98.9 % (Figure. 3h).

The authors are using CA 15-3 (MUC1) as a control of soluble glycoproteins with ConA. ConA does not bind mucin-type O-glycans, and even though MUC1 is moderately N-glycosylated, the authors have not shown that ConA does bind to their antigen used as a control.

Line 383 “Although ultracentrifugation has been widely adopted as the gold standard for isolating EVs from plasma samples, it suffers from co-precipitation of protein aggregates with EVs and relatively low

and unsteady recovery of EVs that substantially affect the downstream detection of EV glycan.” As mentioned earlier, ultracentrifugation is not the gold standard for clinical analysis of EVs.

The clinical cohort in Figure 6 is small. As a discovery cohort this could be sufficient. However the authors do not include a standard confirmation cohort to validate their model.

Flow cytometry on cells treated with cisplatin is not sufficiently explained. Do the authors gate on viable cells? I.e. was a viability dye used? Given this information is not in the methods it would appear that no viability dye is used. As a result, there could be a significant impact on the flow cytometry results due to dead cells.

The authors claim the cohorts are aged matched with each cohort age as 52 (35 –80) 52 (35 – 70) 45 (23 – 64). There seems to be a significant difference in the healthy donors. Could the authors at a minimum provide a statistical test showing no difference.

It is unclear what advantage the proposed thermophoretic enrichment provides. The authors do not include a control of measuring the fluorescence of the sample from the post-vibrating membrane filtration system, thereby confirming their assay is providing an improvement to sensitivity of detection.

The authors used an ELISA to compare their results in Figure 4 to show improved sensitivity. However, this ELISA uses CD63 as a pull down. The authors have not demonstrated that their EVs are positive for CD63 (or any conventional EV marker). As a result, it is not known if the ELISA would work. Furthermore, due to the heterogeneity of EVs, not all EVs are CD63+, and as a result, only a subset of EVs would be analysed using ELISA. Therefore, the sensitivity comparison EVs using thermophoretic enrichment compared to CD63+ EVs using ELISA is not appropriate or a valid comparison.

The discussion is just a reiteration of the results and shows little insight, and in general could be improved. This is also pertinent as to why EV glycosylation is important in predicting treatment outcomes in patients.

Line 558 “The samples at the microchamber center were locally heated by a 1480-nm laser (Changchun Laser Optoelectronics Technology, China) with a power of 150 mW for 10 min for thermophoretic enrichment of EVs.” Although the authors provide a model for the proposed mechanism in Figure 4a, there is insufficient evidence as to what temperature is achieved and how this drives size-dependent accumulation of particles around the bottom of the microchamber. What is the minimum/maximum size of particles, what temperature is required etc.

Authors' Response to the Review Comments

We are grateful to the reviewers for their constructive comments and insightful suggestions. Below, the reviewers' comments are shown in *blue*, and our responses are in black. Sections marked by *red* represent all altered contents in the revised manuscript. The following are our detailed responses.

Response to Reviewer #1

Comment 1: This study applies a lectin microarray to classify EVs in the context of TNBC diagnostics. The authors have adapted a conventional microarray to utilize thermophoretic capabilities. Overall, the idea is pretty compelling, and even tested in >100 clinical samples. I have a few clarifications and suggestions related to perceived weaknesses of EV enrichment and characterization.

Response: We thank the reviewer for the positive evaluation of our research work and providing thoughtful and insightful comments, which have helped to improve the quality of our manuscript. The following comments have been thoroughly addressed upon the reviewer's request.

Comment 2: Please comment on the pore size of the membrane filtration overlapping with EV size. The mode of EV diameter is in the range of 50-60 nm, with some/many as small as 30 – 40 nm. These are not perfectly rigid spheres and so some small EVs could be lost through the filter. Also LDL (20 – 30 nm) and VLDL (30-90 nm) are heavily overlapping right in the size range of the EVs and larger than the filter pore sizes. About ~20% of the pores are larger than 40 nm, presumably a "soft" 60-70 nm EV could be compressed through here? How could EVs be retained while the LDL/VLDL goes through? Obviously some larger EVs are retained as shown in the excellent SEM images, but then you are only examining the larger ones (>100 nm). This could be fine if the lectins on these larger EVs are the ones that carry the diagnostic information but at the least this point should be carefully clarified in the manuscript. What about chylomicrons - other LPPs that are highly abundant and glycosylated? These are seemingly ignored. How exactly are the recovery rates calculated? Just by the free FITC-ConA? How do you know it's free and not small-EV-bound? Also this wouldn't tell you how much non-ConA labeled EVs got through. What is the distribution of lectins across all EVs? Hard to believe this is a universal label.

Response: We appreciate the reviewer's insightful comment. Previous studies showed

that about 80 % of plasma-derived EVs are larger than 50 nm and the mode of EV size is 50 – 100 nm (Supplementary Table 2). The majority of cell line derived-EVs are also larger than 50 nm. In comparison, the size of low-density lipoprotein (LDL) is ranged between 18 – 25 nm, and that of very-low density lipoprotein (VLDL) is from 30 to 80 nm with over 90 % of VLDL smaller than 50 nm^{1, 2}. Given that 20 % of anodized aluminum oxide (AAO) membrane pores are 40 – 50 nm in size, the vast majority of LDL and VLDL may pass through the large pores of AAO membrane, while EVs could be retained on the membrane. By measuring the LDL and VLDL concentrations before and after the vibrating membrane filtration (VMF), we showed that > 99.1 % of LDL and > 98.9 % of VLDL could be removed. As to the retention of EVs, the wide-field TEM and NTA characterization showed that the size distributions of MDA-MB-231 EVs before and after VMF were almost consistent (Supplementary Figs. 9-10). In addition, the percentage of MDA-MB-231 EVs smaller than 100 nm was 27 %, which was slightly decreased to 24 % after VMF. SEM imaging also verified the effective isolation of plasma EVs that are smaller than 100 nm by VMF (Supplementary Fig. 11). Cumulatively, these investigations confirm the capability of VMF for isolating and purifying EVs in clinical samples.

To assess whether the diagnostic information was carried by large EVs, we used a hydrophobic substrate to adsorb cell line-derived EVs (both large and small EVs) for lectin-based ELISA, and compared the results with those obtained by EVLET (large EVs after the filtration). As shown in Fig. 5c and Supplementary Fig. 17, we found strong correlations between EVLET and lectin-based ELISA (Pearson correlation coefficient = 0.90) for measuring the glycan patterns of MDA-MB-231, MDA-MB-453, BT-474, MCF-7, and MCF-10A EVs using a panel of lectins (ConA, WGA, or RCA I). This suggests that EVLET can comprehensively detect EV glycan profiles despite the loss of a tiny amount of small EVs.

We agree with the reviewer that chylomicrons (CM), with a size range of 75 – 1200 nm, can be co-isolated with EVs by VMF. However, since plasma samples were collected from donors after overnight fasting, most of CM has been cleared from the circulation due to its short half-life time (10 – 20 min)^{3, 4}. To assess the effect of lipoproteins on EV glycan analysis, we applied EVLET to detect the healthy donor (HD) plasma sample spiked with LDL (2.7 mg mL⁻¹)⁵ or CM (1.3 mg mL⁻¹)^{3, 4, 6}. As shown in Supplementary Fig. 16, there was no significant difference in signal intensity between the HD sample and the spiked sample, indicating that lipoproteins have a minimal effect on EV glycan detection.

The recovery rate of EVs was defined as the ratio of the number of EVs after VMF to that of before, which was determined by NTA measurement. Without the presence of EVs, a high removal rate (up to 99.9 %) of free FITC-ConA was achieved by VMF,

indicating that the unbound FITC-ConA would not affect the downstream detection of EV glycan profiles (Supplementary Fig. 12). However, the percent of ConA- or non-ConA-labeled EVs through the AAO membrane remained unknown. To evaluate the distribution of lectins across EVs, we used ConA for labeling EV glycans and lipophilic DiD for EV membranes. Fluorescence analysis revealed that 25.8 % of MDA-MB-231 EVs and 6.6 % of MCF-10A EVs had DiD-FITC colocalization, implying a heterogeneous distribution of lectins across EVs (Supplementary Fig. 13). We have added these experimental data in the revised manuscript as follows.

Results

Design of EVLET

EVLET integrated a customized VMF for EV isolation and a thermophoretic assay for EV glycan detection. We first characterized the performance of VMF for automatic filtration, washing, and retrieval of EVs secreted by the TNBC-derived MDA-MB-231 cell line (Fig. 3a and Supplementary Fig. 1). VMF consisted of a nanoporous anodic aluminum oxide (AAO) membrane and a high-frequency oscillator (6 kHz, 25 V). The oscillator-enabled vibration of AAO membrane periodically lifted up the MDA-MB-231 EVs to prevent membrane fouling⁷. **Despite the nominal pore size of 20 nm, scanning electron microscopy (SEM) characterization indicated that ~ 20 % of pores on AAO membrane were in the size range of 40 – 50 nm (Supplementary Fig. 3). Given that the majority of cell line derived-EVs were larger than 50 nm, MDA-MB-231 EVs could be retained on AAO membrane after VMF (Supplementary Table 2). As seen in SEM images, the isolated MDA-MB-231 EVs remained round-shaped with no detectable damage, while EVs smaller than 100 nm were also observed (Fig. 3b and Supplementary Fig. 6). The wide-field transmission electron microscope (TEM) and nanoparticle tracking analysis (NTA) further confirmed that the size distributions of MDA-MB-231 EVs were almost consistent before and after VMF (Supplementary Figs. 7-10). The percentage of MDA-MB-231 EVs smaller than 100 nm was slightly decreased from 27 % to 24 % after VMF. The recovery rate of MDA-MB-231 EVs by VMF (6 kHz) was 22.2 %, 1.9-fold higher than that without vibration (0 kHz, Fig. 3c). VMF showed similar recovery rates ranged from 20.1 % – 22.2 % for EVs from 4 other breast cell lines (MDA-MB-453, MCF-7, BT-474, and MCF-10A), with a low coefficient of variation of 5.2 % – 9.5 % (Fig. 3d and Supplementary Fig. 8).**

Supplementary Fig. 6 | SEM images of MDA-MB-231 EVs on AAO membranes after VMF. 200 μL of MDA-MB-231 EVs ($10^7 \mu\text{L}^{-1}$) was loaded onto the membrane filter. EVs smaller than 100 nm were indicated by yellow arrows, EVs larger than 100 nm were indicated by red arrows. Scale bar, 200 nm.

Supplementary Fig. 9 | TEM images of MDA-MB-231 EVs before and after VMF. 200 μL of MDA-MB-231 EVs ($10^7 \mu\text{L}^{-1}$) was loaded onto the membrane filter. Scale bar, 500 nm.

Supplementary Fig. 10 | Size distribution of MDA-MB-231 EVs. TEM (bar) and NTA (line) showing the size distribution of MDA-MB-231 EVs before and after VMF. TEM characterization indicating that 27 % of EVs before VMF and 24 % of EVs after VMF were smaller than 100 nm ($n = 284$ EVs before VMF, $n = 255$ EVs after VMF).

Supplementary Table 2 | Size distribution of EVs.

EV source	Isolation method	Size measurement	Size distribution
MDA-MB-231 cell	UC	Cryo-EM	95 % EVs > 50 nm, mode size of 120 nm ⁸
B16-F1 cell	UC	Cryo-EM	Almost all EVs > 50 nm, mode size of 100 nm ⁹
Jurkat, THP-1, U937, MiaPaCa cell lines	UC	TRPS	All EVs: > 60 nm, mode sizes: ~ 100 nm ¹⁰
BT-474 cell	UC	TRPS	All EVs > 50 nm, mode size of 60 – 70 nm ¹¹
Human plasma	UC	Cryo-EM	Over 80 % EVs > 50 nm ¹²

Human plasma	NA	Cryo-EM	Over 80 % EVs > 100 nm, mode size: 100 – 200 nm ¹³
Human plasma	UC	TRPS	Almost all EVs > 50 nm, mode size ~ 100 nm ¹⁴
Human plasma	SEC	TRPS	mode size: 55 – 73 nm ¹⁵

UC: ultracentrifugation

SEC: size exclusion chromatography

TRPS: tunable resistive pulse sensing

We further applied VMF to isolate and purify EVs from plasma samples that contain large amounts of circulating proteins, low-density lipoproteins (LDL, 18 – 25 nm), and very low-density lipoproteins (VLDL, 30 – 80 nm with over 90 % of VLDL smaller than 50 nm)^{1,2} (Supplementary Fig. 11). Benefiting from the large pores (40 – 50 nm) of AAO membranes, 99.5 % of proteins, > 99.1 % of LDL, and > 98.9 % of VLDL in plasma could be removed by VMF (details in methods, Supplementary Table 3).

Supplementary Fig. 11 | SEM images of plasma EVs on AAO membranes after VMF. Human plasma (2 μ L, diluted by 750 folds) was loaded onto the membrane filter. EVs smaller than 100 nm were indicated by yellow arrows, EVs larger than 100 nm were indicated by red arrows. Scale bar, 200 nm.

Supplementary Table 3 | Comparison between ultracentrifugation (UC), size exclusion chromatography (SEC), tangential flow filtration (TFF), and vibrating membrane filtration (VMF) in processing plasma/serum samples.

	UC	SEC	TFF	VMF (present study)
--	----	-----	-----	------------------------

Recovery rate	1 % (present) < 5 % ^{16, 17}	39.3 % (present) 60 % – 85 % ^{18, 19}	60 % – 80 % ²⁰⁻²²	22.1 % (present)
Purity (particles/μg protein)	1.8×10^7 (present) $10^7 - 2 \times 10^8$ ^{20, 23, 24}	1.3×10^9 (present) $0.5 \times 10^9 - 6 \times 10^9$ ²⁴⁻²⁶	$10^7 - 1.2 \times 10^8$ ^{20, 22}	9.3×10^8 (present)
Removal rate of contaminants	Total protein: 99.3 % LDL: 96.7 % VLDL: 98.6 % (present)	Total protein: 99.4 % LDL: 95.7 % VLDL: 88.7 % (present)	-	Total protein: 99.5 % LDL: > 99.1 % VLDL: > 98.9 % (present)
Sample volume	2 mL (present) 1 mL – 4 mL ^{23, 24}	2 mL (present) 0.15 mL – 4 mL ^{23, 25,} ²⁷⁻²⁹	0.03 – 0.6 mL ²⁰⁻²²	0.002 mL (present)
Processing time	3 h (present) 1.4 – 3.5 h ^{23, 25, 30, 31}	3.5 h (present) 1.5 – 4 h ^{30, 32}	< 0.7 – 3 h ²⁰⁻²²	0.16 h (present)

Next, we adopted a thermophoretic assay for sensitive detection of EV glycans that were labeled by fluorescent lectins (Fig. 4). As unbound lectins may affect the detection accuracy, VMF was used to remove free lectins of small size. By measuring the fluorescence intensity of FITC-ConA (without the presence of EVs) before and after VMF, we found that 99.9 % of FITC-ConA could be filtered out by VMF (Supplementary Fig. 12). To evaluate whether lectin is a universal label for EV glycans, we used FITC-ConA for labeling EVs and lipophilic DiD for EV membranes. Fluorescence analysis revealed that 25.8 % of MDA-MB-231 EVs and 6.6 % of MCF-10A EVs (benign breast cell line-derived EVs) had DiD-FITC colocalization after VMF, suggesting a heterogeneous distribution of lectins across EVs and a higher expression of glycans on TNBC cell line-derived EVs (Supplementary Fig. 13).

Supplementary Fig. 12 | Quantification of removal rate of free FITC-ConA

(without the presence of EVs) by VMF. **a**, Standard curve for calculating the removal rate of free lectin after VMF. The linear range of detection of ConA by fluorescence spectrometer was $0.008 - 25 \mu\text{g mL}^{-1}$ ($n = 3$, mean \pm s.d.). **b**, The concentration of free FITC-ConA before or after VMF. $5 \mu\text{g mL}^{-1}$ ConA solution was used as the input and the concentration of ConA in the filtrate was quantified based on the standard curve ($n = 3$, mean \pm s.d.).

Supplementary Fig. 13 | Fluorescence colocalization of DiD- and Con A (FITC)-labeled EVs. MDA-MB-231 EVs and MCF-10A EVs were subjected to ConA labeling, VMF, and DiD labeling before fluorescence microscopy observation. Scale bar, 20 μm .

EVLET was also applied to detect HD plasma sample mixed with LDL (2.7 mg mL^{-1})⁵ or chylomicrons (CM, 1.3 mg mL^{-1})^{3, 4, 6}. As shown in Supplementary Fig. 16, there was no significant difference in signal intensity between the HD sample and the spiked sample, indicating that lipoproteins have a minimal effect on EV glycan detection. Collectively, our EVLET system enabled sensitive, specific, and quantitative analysis of EV glycans.

Supplementary Fig. 16 | Minimal effect of lipoproteins on EV glycan detection. Fluorescence intensities of 8 types of samples measured by EVLET: (1) HD plasma spiked with MDA-MB-231 EVs (1.4×10^8 EVs per assay) and LDL (2.7 mg mL^{-1}); (2) HD plasma spiked with MDA-MB-231 EVs and CM (1.3 mg mL^{-1}); (3) HD plasma spiked with MDA-MB-231 EVs; (4) HD plasma spiked with LDL; (5) HD plasma spiked with CM; (6) HD plasma; (7) LDL; (8) CM ($n = 3$, mean \pm s.d.).

EVLET for profiling EV glycans

We next sought to profile the expression of glycans on EVs derived from different breast cell lines (MDA-MB-231, MDA-MB-453, BT-474, MCF-7, and MCF-10A) by EVLET using a panel of lectins (ConA, WGA, RCA I, SBA, and UEA I). Fig. 5a-b showed that EV glycan patterns were varied across different types of EVs, which were also confirmed by lectin-based ELISA (Pearson correlation coefficient $r = 0.90$, $p < 0.0001$, Fig. 5c and Supplementary Fig. 17).

Fig. 5 | c, Good correlation between the glycan pattern of EVs measured by EVLET and lectin-based ELISA ($n = 3$, mean \pm s.d.).

Supplementary Fig. 17 | Glycan patterns of cell line EVs. EV glycans measured by lectin-based ELISA (top row) and EVLET (bottom row) using ConA, WGA, and RCA I ($n = 3$, mean \pm s.d.).

Methods

Removal of plasma proteins and lipoproteins by different methods

To evaluate the removal efficiency of proteins, plasma samples were first processed by VMF (2 μ L plasma per sample), UC (2 mL plasma per sample), or SEC (2 mL plasma per sample) following the established protocol. The recovered samples and unprocessed plasma samples were analyzed using BCA Protein Assay Kit. Briefly, 20 μ L of samples were pipetted into microplate wells and mixed with 200 μ L of BCA working reagent. After incubation for 30 min at 37 $^{\circ}$ C, the optical density (OD) of the solution at 562 nm was immediately detected using a microplate reader (Synergy H1, Biotek, USA). The protein concentrations of the recovered samples and unprocessed plasma samples were calculated based on the standard curve. The EV purity was determined as the number of particles per μ g protein.

To evaluate the removal efficiency of lipoproteins (LDL and VLDL), plasma samples were first processed by VMF (2 μ L plasma per sample), UC (2 mL plasma per sample), or SEC (2 mL plasma per sample) following the established protocol. The recovered samples and unprocessed plasma samples were analyzed using LDL or VLDL ELISA kits. Briefly, the samples were adsorbed onto ELISA plates coated with

anti-LDL or anti-VLDL antibody for 1 h at 37 °C (100 µL for each well). The supernatant was discarded, and the plates were incubated with 100 µL of biotin-conjugated LDL or VLDL antibody for 1 h at 37 °C. After washing with washing buffer for 3 times, 100 µL of SA-conjugated HRP was added and incubated for 30 min at 37 °C. The supernatant was discarded again, and ELISA plates were washed with washing buffer for 5 times. 90 µL of TMB substrate solution was added and incubated for 15 min at 37 °C. The reaction was terminated by the addition of 50 µL of stop solution and the OD of the solution at 450 nm was immediately detected using a microplate reader. The lipoprotein concentrations of the recovered samples and unprocessed plasma samples were calculated based on the standard curve.

Distribution of ConA across EVs

To determine the distribution of lectins across EVs, FITC-ConA (5 µg mL⁻¹) was used to label MDA-MB-231 EVs and MCF-10A EVs (10⁷ µL⁻¹, 200 µL) at room temperature for 1 h. The EV samples were then filtered through VMF (6 kHz, 25 V, 50 kPa), washed twice with 150 µL PBS, and retrieved in 100 µL PBS. The recovered samples were treated with 4 nM DiD to label EVs at 37 °C for 20 min. The fluorescence colocalization of DiD and FITC signals was analyzed under a fluorescence microscope (DMi8, Leica, Germany) with a 100 × objective, and the images were captured with an sCMOS camera (95B, Photometrics, Canada) using 2 × 2-pixel binning and a 50 ms exposure time. For MDA-MB-231 EVs, the percent of ConA-labeled EVs was 25.8 % based on 90 DiD-FITC colocalization events out of 349 EVs (DiD). For MCF-10A EVs, the percent of ConA-labeled EVs was 6.6 % based on 22 DiD-FITC colocalization events from 332 EVs (DiD).

Comment 3: *The characterization of the EVs retained by the custom vibrating membrane filtration system is very weak. Only SEM (with a few EVs) and NTA (known to be very inaccurate to size and assess concentration of EVs) is performed. NTA can only see the >80 nm particles which represent less than 30% of the total number at most. The authors should refer the MISEV-2018 document which provides a comprehensive list of suggested assays that minimally is needed for accurate reporting that EVs are recovered in high yield. This is especially critical for a "new" type of EV isolation reported here. Don't get me wrong - it's a very exciting technology with great application, just need some more details on the characterization of the EVs emerging from this unit. I strongly suggest performing more accurate EV sizing in the relevant <100 nm range like resistive pulse sensing, cryo-electron microscopy (not on filter but after collection from the filter), single particle interferometric imaging, or super resolution microscopy. These other methods would be critical to support the claims.*

Response: We appreciate reviewer for the valuable suggestion, which significantly enhanced the rigor of our study. To characterize the performance of NTA (NanoSight 300), we first adjusted the particle concentration in the range of $2 \times 10^5 \mu\text{L}^{-1} - 8 \times 10^5 \mu\text{L}^{-1}$ to circumvent the artifacts arising from the overlap of particle trajectories. Moreover, a high camera level and a low threshold value were set to avoid underestimate in particle concentration³³. Under the optimized conditions, polystyrene particles of different sizes (41 nm, 47 nm, and 100 nm) can be accurately detected by NTA (Supplementary Fig. 7a). Different concentrations of 47 nm particles were also precisely quantified by NTA with deviations less than 12.7 % (Supplementary Fig. 7b). In response to the reviewer's suggestion, we also employed resistive pulse sensing (RPS) to characterize the plasma EVs before and after VMF. The size distributions, mode size, and concentrations of plasma EVs measured by RPS and NTA showed a close agreement, affirming the reliability of NTA to size and assess concentration of EVs (Response Fig. 1).

Furthermore, a comprehensive list of assays suggested by the MISEV-2018 document was carried out to characterize the recovery and purity of EVs isolated by VMF. NTA was used to measure the particle concentrations in plasma samples before and after VMF, while BCA assay was used to detect the total protein content. As summarized in Supplementary Table 3, VMF yielded a recovery rate (22.1 %) that was substantially higher than UC (1 %), while being lower than SEC (39.3 %) and TFF (60 % – 80 %). Additionally, both VMF and SEC exhibited the high purity (9.3×10^8 particles per μg protein for VMF and 1.3×10^9 for SEC), which was significantly higher than UC (1.8×10^7) and TFF ($10^7 - 1.2 \times 10^8$). Compared to the other three methods, VMF is compatible for small sample volumes (down to 2 μL of plasma) and is fully automatic with a total processing time around 10 min (Supplementary Table 3). We also performed western blotting (WB) to analyze the expression of specific proteins of plasma EVs after VMF. Results of WB indicated the enrichment of EV markers, CD81 and HSP90, and the absence of non-EV markers, calnexin and ApoB, verifying the high-purity of VMF-isolated EVs (Fig. 3g). These experimental data have now been included in the revised manuscript.

Supplementary Fig. 7 | Size detection limit of NTA. **a**, NTA measurement of size distribution of polystyrene (PS) particles of nominal diameters of 41 nm, 47 nm, and 100 nm, and a mixture of 47 nm and 100 nm particles. **b**, NTA measurement of 47 nm particles with varied concentrations ($n = 3$, mean \pm s.d.).

Response Fig. 1 | Performance of RPS. **a**, Size distribution of PS particles of nominal diameters of 100 nm measured by RPS. **b**, RPS measurement of 100 nm particles with varied concentrations.

Response Fig. 2 | Characterization of size distribution and concentration of bioparticles by RPS and NTA. a, Similar size distributions for a plasma sample before and after VMF as determined by NTA (lines) and RPS (bars). **b**, Particle concentrations before and after VMF measured by NTA and RPS ($n = 3$, mean \pm s.d.). **c**, Recovery rates calculated based on the concentrations in (b) ($n = 3$, mean \pm s.d.).

Results

Design of EVLET

To determine the recovery and purity of EVs, a comprehensive list of assays suggested by the MISEV-2018 document was carried out to characterize EVs isolated by VMF and other methods¹⁷⁻²⁰. As summarized in Supplementary Table 3, VMF yielded a recovery rate (22.1 %) that was substantially higher than ultracentrifugation (UC, 1 %), while being lower than size exclusion chromatography (SEC, 39.3 %) and tangential flow filtration (TFF, 60 % – 80 %). Both VMF and SEC exhibited the high purity (9.3×10^8 particles per μg protein for VMF and 1.3×10^9 for SEC), which was better than UC (1.8×10^7) and TFF ($10^7 - 1.2 \times 10^8$) (Fig. 3e). The removal rates of total proteins, LDL, and VLDL by VMF were higher than UC and SEC (Fig. 3f). Western blotting

(WB) of plasma EVs isolated by VMF showed the enrichment of EV markers, CD81 and HSP90, and the absence of non-EV markers, calnexin and ApoB, verifying the high-purity of plasma EVs after VMF (Fig. 3g). Compared to the other three methods, VMF is compatible for small sample volumes (down to 2 μ L of plasma) and is fully automatic with a total processing time around 10 min (Supplementary Table 3), providing a powerful tool for EV isolation and purification.

Fig. 3 | e, Comparison of purity of EVs isolated from plasma samples by VMF, UC, or SEC ($n = 3$, mean \pm s.d.). **f**, Comparison of removal rates of total proteins, LDL, and VLDL in plasma samples using VMF, UC, or SEC ($n = 3$, mean \pm s.d.). **g**, Western blot analysis of recovered samples from plasma by VMF, UC, or SEC. The samples were loaded with equal protein amounts.

Supplementary Table 3 | Comparison between ultracentrifugation (UC), size exclusion chromatography (SEC), tangential flow filtration (TFF), and vibrating membrane filtration (VMF) in processing plasma/serum samples.

	UC	SEC	TFF	VMF (present study)
Recovery rate	1 % (present) < 5 % ^{16, 17}	39.3 % (present) 60 % – 85 % ^{18, 19}	60 % – 80 % ²⁰⁻²²	22.1 % (present)
Purity (particles/μg protein)	1.8×10^7 (present) $10^7 - 2 \times 10^8$ ^{20, 23, 24}	1.3×10^9 (present) $0.5 \times 10^9 - 6 \times 10^9$ ²⁴⁻²⁶	$10^7 - 1.2 \times 10^8$ ^{20, 22}	9.3×10^8 (present)
Removal rate of contaminants	Total protein: 99.3 % LDL: 96.7 % VLDL: 98.6 % (present)	Total protein: 99.4 % LDL: 95.7 % VLDL: 88.7 % (present)	-	Total protein: 99.5 % LDL: > 99.1 % VLDL: > 98.9 % (present)

Sample volume	2 mL (present) 1 mL – 4 mL ^{23, 24}	2 mL (present) 0.15 mL – 4 mL ^{23, 25,} 27-29	0.03 – 0.6 mL 20-22	0.002 mL (present)
Processing time	3 h (present) 1.4 – 3.5 h ^{23, 25, 30, 31}	3.5 h (present) 1.5 – 4 h ^{30, 32}	< 0.7 – 3 h ²⁰⁻²²	0.16 h (present)

Methods

Nanoparticle tracking analysis (NTA)

To characterize the size distribution and concentration, samples containing EVs were detected using NTA (NanoSight NS300, Malvern Instrument, England) at 23 ± 2 °C. Particle concentrations were adjusted to $2 \times 10^5 - 8 \times 10^5$ particles μL^{-1} to circumvent the artifacts arising from the overlap of particle trajectories. A high camera level (15, maximum = 16) and a low threshold value (5, maximum = 50) was set to avoid underestimation in particle concentration. The data of size distribution were captured and analyzed with the NTA 3.4 Analytical Software Suite. To evaluate the performance of NTA, polystyrene (PS) particles (Invitrogen, USA) of three different nominal sizes, 41 nm, 47 nm, and 100 nm, were subjected to NTA measurement. The sizes of these particles were accurately identified (Supplementary Fig. 7a). Additionally, the accuracy of concentration quantification was verified by detecting 47 nm particles with varying concentrations ($2 \times 10^5 \mu\text{L}^{-1} - 8 \times 10^5 \mu\text{L}^{-1}$). The concentrations measured by NTA were aligned well with the actual concentrations ($R^2 = 0.9160$) (Supplementary Fig. 7b).

Western blot (WB)

The plasma samples were processed by VMF (2 μL plasma for one run of filtration, and the recovered samples were concentrated to meet the protein loading amount for WB analysis), ultracentrifugation (2 mL plasma per sample), or SEC (2 mL plasma per sample) following the established protocol. The recovered samples were lysed by 1 \times SDS-PAGE loading buffer at 100 °C for 10 min. Samples and standard ladder were loaded onto a gel with equal protein amounts (1.5 μg), separated by electrophoresis (80 V for 30 min and 140 V for 60 min), and transferred onto a PVDF membrane. The PVDF membrane was blocked with TBST buffer (tris-buffered saline and 0.1% Tween 20) containing 5 % BSA for 30 min at room temperature. The blocked PVDF membrane was incubated with anti-CD81 antibody (1/1000), anti-HSP90 antibody (1/500), anti-Calnexin antibody (1/500), and anti-ApoB antibody (1/10000) at 4 °C overnight, followed by washing three times for 10 min using TBST buffer. The PVDF membrane was then incubated with HRP-conjugated anti-rabbit IgG (1/5000) at room temperature

for 1 h, followed by washing three times for 10 min using TBST buffer. The protein bands were visualized using chemiluminescence reagent and observed on a Tanon 5200 multi-gel analysis (Tanon Shanghai, China).

Response to Reviewer #2

Comment 1: *In this study, Li et al. sought to design a lectin based (EVLET) platform for diagnosis and monitoring of TNBC patients. In general, the manuscript is well written and presented, and the inclusion of longitudinal sampling is one of the strongest points in this manuscript. The concept of thermophoretic enrichment of EVs is interesting and innovative, but lacks novelty as this group has previously published this (PMID: 32134270). Furthermore, the field has moved a long way from when ultracentrifugation was the gold standard. It is now commonplace to use either/both tangential flow filtration, and size exclusion chromatography. The authors do not take this into account throughout the manuscript. As a result, there is no comparison to commonly used techniques, and is therefore not possible to draw conclusions on the performance of the purification strategy. As a result of this, as well as some more specific comments below, I do not recommend this for publication.*

Response: We truly appreciate reviewer for the in-depth reading as well as detailed comments and suggestions. Previously, we successfully achieved EV miRNA analysis for early-stage breast cancer diagnosis by the combination of nanoflakes and thermophoretic assay (PMID: 32134270). However, costly and time-consuming ultracentrifugation procedure for EV isolation impedes the clinical use of this method. Moreover, the co-precipitation of protein aggregates with EVs by ultracentrifugation may affect the measurement of EV glycans. In current study, the most innovative aspect is that we devise a clinically feasible platform (termed as EVLET) to analyze EV glycan profiles in the raw plasma without lengthy isolation procedures and complex detection steps, yet to achieve high sensitivity and specificity. The potential of EV glycan profiles measured by EVLET is further explored for non-invasive management of TNBC, an aggressive type of breast cancer with poor outcomes. In our cohort study involving more than 100 clinical samples, the EV glycan signature shows a high accuracy for diagnosis, therapeutic response monitoring, and prognosis of TNBC.

We agree with the reviewer in that the field has moved a long way from when ultracentrifugation (UC) was the gold standard. Nevertheless, recent worldwide survey conducted by the International Society for Extracellular Vesicles (ISEV)³⁴ shows that UC remains the most commonly used method for EV isolation, while size exclusion chromatography (SEC) and tangential flow filtration (TFF) are less frequently employed. Per the reviewer's request, we compare our vibrating membrane filtration (VMF) with UC, SEC, and TFF for isolating EVs in plasma samples. As summarized in Supplementary Table 3, VMF yielded a recovery rate (22.1 %) that was substantially higher than UC (1 %), while being lower than SEC (39.3 %) and TFF (60 % – 80 %).

Additionally, both VMF and SEC exhibited the high purity (9.3×10^8 particles per μg protein for VMF and 1.3×10^9 for SEC), which was significantly higher than UC (1.8×10^7) and TFF ($10^7 - 1.2 \times 10^8$). Compared to the other three methods, VMF is compatible for small sample volumes (down to 2 μL of plasma) and is fully automatic with a total processing time around 10 min.

From these aspects, we believe that our EVLET platform can provide a significant advance as a diagnostic toolbox for EV glycan analysis and is of promising potential for TNBC management.

Results

Design of EVLET

To determine the recovery and purity of EVs, a comprehensive list of assays suggested by the MISEV-2018 document was carried out to characterize EVs isolated by VMF and other methods¹⁷⁻²⁰. As summarized in Supplementary Table 3, VMF yielded a recovery rate (22.1 %) that was substantially higher than ultracentrifugation (UC, 1 %), while being lower than size exclusion chromatography (SEC, 39.3 %) and tangential flow filtration (TFF, 60 % – 80 %). Both VMF and SEC exhibited the high purity (9.3×10^8 particles per μg protein for VMF and 1.3×10^9 for SEC), which was better than UC (1.8×10^7) and TFF ($10^7 - 1.2 \times 10^8$) (Fig. 3e and Supplementary Table 3). The removal rates of total proteins, LDL, and VLDL by VMF were higher than UC and SEC (Fig. 3f). Western blotting (WB) of plasma EVs isolated by VMF showed the enrichment of EV markers, CD81 and HSP90, and the absence of non-EV markers, calnexin and ApoB, verifying the high-purity of plasma EVs after VMF (Fig. 3g). Compared to the other three methods, VMF is compatible for small sample volumes (down to 2 μL of plasma) and is fully automatic with a total processing time around 10 min (Supplementary Table 3), providing a powerful tool for EV isolation and purification.

Fig. 3 | **e**, Comparison of purity of EVs isolated from plasma samples by VMF, UC, or SEC ($n = 3$, mean \pm s.d.). **f**, Comparison of removal rates of total proteins, LDL, and

VLDL in plasma samples using VMF, UC, or SEC ($n = 3$, mean \pm s.d.). **g**, Western blot analysis of recovered samples from plasma by VMF, UC, or SEC. The samples were loaded with equal protein amounts.

Supplementary Table 3 | Comparison between ultracentrifugation (UC), size exclusion chromatography (SEC), tangential flow filtration (TFF), and vibrating membrane filtration (VMF) in processing plasma/serum samples.

	UC	SEC	TFF	VMF (present study)
Recovery rate	1 % (present) < 5 % ^{16, 17}	39.3 % (present) 60 % – 85 % ^{18, 19}	60 % – 80 % ²⁰⁻²²	22.1 % (present)
Purity (particles/μg protein)	1.8×10^7 (present) $10^7 - 2 \times 10^8$ ^{20, 23, 24}	1.3×10^9 (present) $0.5 \times 10^9 - 6 \times 10^9$ ²⁴⁻²⁶	$10^7 - 1.2 \times 10^8$ ^{20, 22}	9.3×10^8 (present)
Removal rate of contaminants	Total protein: 99.3 % LDL: 96.7 % VLDL: 98.6 % (present)	Total protein: 99.4 % LDL: 95.7 % VLDL: 88.7 % (present)	-	Total protein: 99.5 % LDL: > 99.1 % VLDL: > 98.9 % (present)
Sample volume	2 mL (present) 1 mL – 4 mL ^{23, 24}	2 mL (present) 0.15 mL – 4 mL ^{23, 25,} ²⁷⁻²⁹	0.03 – 0.6 mL ²⁰⁻²²	0.002 mL (present)
Processing time	3 h (present) 1.4 – 3.5 h ^{23, 25, 30, 31}	3.5 h (present) 1.5 – 4 h ^{30, 32}	< 0.7 – 3 h ²⁰⁻²²	0.16 h (present)

Methods

Size exclusion chromatography (SEC)

SEC was used to process the plasma samples. SEC column (qEV2, 35 nm) was flushed by 45 mL PBS before use. 2 mL plasma was directly loaded into the column. The loaded sample was eluted with PBS, and EV-containing fractions (14 – 22 mL) were pooled. For reuse, the column post plasma processing was sequentially washed with 90 mL of PBS, 2 mL of 0.5 M NaOH, and 90 mL of PBS to remove the residual proteins.

EV isolation by ultracentrifugation

For plasma EV isolation, the plasma samples (2 mL) were diluted to 27 mL using $1 \times$ PBS and centrifuged at $2,000 \times g$ for 10 min by twice. The resulting supernatant was filtered through a $0.45 \mu\text{m}$ Millipore filter and ultracentrifuged at $100,000 \times g$ for 3 h.

The EVs were obtained by suspending the pellet in 200 μ L of 1 \times PBS. Before ultracentrifugation, the weight difference between every pair of tubes was set to be smaller than 0.05 g. The collected cell line EVs and plasma EVs were stored at -80 $^{\circ}$ C before use.

Comment 2: *Several sentences throughout the manuscript make claims without providing evidence/relevant citation. For example, Line 55 “Despite its feasibility, lectin microarray has not yet reached sufficient sensitivity due to the diffusion-limited mass transport of EVs to the substrate.” If this has been previously shown, please cite the relevant literature.*

Response: We thank the reviewer for these detailed comments. The relevant literatures indicating the insufficient sensitivity of lectin microarray have been added as follows.

Introduction

In spite of its feasibility, the sensitivity of lectin microarray could be compromised due to the diffusion-limited mass transport of EVs³⁵.

Comment 3: *Line 59 “These microfluidic methods involve the fabrication of complex microfluidic devices or functional nanoparticles, thus hampering their clinical applications.” How does this hamper clinical applications?*

Response: We thank the reviewer for this inquiry. Per the reviewer’s comment, we have revised the description of microfluidic methods for clinical detection of EVs.

Introduction

These microfluidic methods have great potential for clinical detection of EVs. However, the fabrication of microfluidic sensors may require access to clean rooms and specialized skills.

Comment 4: *In terms of the bioinformatic approach to identify differentially expressed proteins involved in glycosylation the authors identify 5525 mRNAs found to be differentially expressed using the TCGA database. The authors then provide evidence that simply the expression of glycosyltransferases or glycosidases can differentiate TNBC. Does this classify TNBC more accurately compared to a model incorporating the 5525 differentially expressed mRNAs? If not, this seems to be redundant as aberrant glycosylation in cancer is a well-established paradigm.*

Response: We appreciate the reviewer’s insightful comment. Per the reviewer’s request, we have applied linear discriminate analysis (LDA) to the TNBC-associated mRNA panel (5525 mRNAs) and glycosylation-related mRNA panel (48 mRNAs) in TCGA-BRCA database (Fig. 2b). The overall accuracy for classifying TNBC, other BC subtypes, and normal tissues by the 5525 TNBC-associated mRNAs was 98.7 %, and that by the 48 glycosylation-related mRNAs was 92.9 %. These results demonstrate the potential of glycan profiles for TNBC diagnosis, which have been added in the revised manuscript as follows.

Results

Identification of TNBC-associated glycans

Linear discriminate analysis (LDA) of 5525 TNBC-associated mRNAs or 48 glycosylation-related mRNAs revealed an overall accuracy of 98.7 % or 92.9 % for classification of TNBC, other BC subtypes, and normal tissues in the TCGA-BRCA cohort ($n = 1153$, Fig. 2b), demonstrating the potential of glycan profiles for TNBC diagnosis.

		TNBC-associated mRNA panel				Glycosylation-related mRNA panel			
		Actual				Actual			
		TNBC	Other BC	Normal	Total	TNBC	Other BC	Normal	Total
Predicted	TNBC	169	4	0	173	147	42	1	190
	Other BC	8	857	1	866	30	813	1	844
	Normal	1	1	112	114	1	7	111	119
		178	862	113	1153	178	862	113	1153
		Accuracy 98.7 %				Accuracy 92.9 %			

Fig. 2 | b, LDA classification of TNBC, other BC subtypes, and normal tissues based on 5525 TNBC-associated mRNAs or 48 glycosylation-related mRNAs.

Comment 5: *The authors are using a vibrating membrane filtration system to purify EVs with a 20 nm nominal pore size. Given the size of LDL > 20 nm and VLDL > 50 nm, how do the authors claim the removal of LDL and VLDL is at > 99.1 %, and > 98.9 %.*

Response: We are thankful for this detailed question. Despite the nominal pore size of 20 nm, SEM characterization of AAO membrane revealed that ~ 20 % of pores were in the size range of 40 – 50 nm (Supplementary Fig. 3). Previous studies demonstrated

that the size of LDL is ranged between 18 – 25 nm, and that of VLDL is from 30 to 80 nm with over 90 % of VLDL smaller than 50 nm^{1,2}. Therefore, most LDL and VLDL may pass through the large pores of AAO membrane. By measuring the LDL and VLDL concentrations before and after VMF, we showed that > 99.1 % of LDL and > 98.9 % of VLDL could be removed. This point has been clarified in the revised manuscript.

Results

Design of EVLET

We further applied VMF to isolate and purify EVs from plasma samples that contain large amounts of circulating proteins, low-density lipoproteins (LDL, 18 – 25 nm), and very low-density lipoproteins (VLDL, 30 – 80 nm with over 90 % of VLDL smaller than 50 nm)^{1,2} (Supplementary Fig. 11). Benefiting from the large pores (40 – 50 nm) of AAO membranes, 99.5 % of proteins, > 99.1 % of LDL, and > 98.9 % of VLDL in plasma could be removed by VMF (details in methods, Supplementary Table 3).

Supplementary Fig. 3 | Characterization of pore size of the AAO membrane. a, SEM image of AAO membrane with a nominal pore diameter of 20 nm. Scale bar, 200 nm. **b,** Size distribution of nanopores in the AAO membrane ($n = 385$ pores) obtained by ImageJ.

Supplementary Fig. 11 | SEM images of plasma EVs on AAO membranes after

VMF. Human plasma (2 μ L, diluted by 750 folds) was loaded onto the membrane filter. EVs smaller than 100 nm were indicated by yellow arrows, EVs larger than 100 nm were indicated by red arrows. Scale bar, 200 nm.

Supplementary Table 3 | Comparison between ultracentrifugation (UC), size exclusion chromatography (SEC), tangential flow filtration (TFF), and vibrating membrane filtration (VMF) in processing plasma/serum samples.

	UC	SEC	TFF	VMF (present study)
Recovery rate	1 % (present) < 5 % ^{16, 17}	39.3 % (present) 60 % – 85 % ^{18, 19}	60 % – 80 % ²⁰⁻²²	22.1 % (present)
Purity (particles/μg protein)	1.8×10^7 (present) $10^7 - 2 \times 10^8$ ^{20, 23, 24}	1.3×10^9 (present) $0.5 \times 10^9 - 6 \times 10^9$ ²⁴⁻²⁶	$10^7 - 1.2 \times 10^8$ ^{20, 22}	9.3×10^8 (present)
Removal rate of contaminants	Total protein: 99.3 % LDL: 96.7 % VLDL: 98.6 % (present)	Total protein: 99.4 % LDL: 95.7 % VLDL: 88.7 % (present)	-	Total protein: 99.5 % LDL: > 99.1 % VLDL: > 98.9 % (present)
Sample volume	2 mL (present) 1 mL – 4 mL ^{23, 24}	2 mL (present) 0.15 mL – 4 mL ^{23, 25,} ²⁷⁻²⁹	0.03 – 0.6 mL ²⁰⁻²²	0.002 mL (present)
Processing time	3 h (present) 1.4 – 3.5 h ^{23, 25, 30, 31}	3.5 h (present) 1.5 – 4 h ^{30, 32}	< 0.7 – 3 h ²⁰⁻²²	0.16 h (present)

Comment 6: *The authors are using CA 15-3 (MUC1) as a control of soluble glycoproteins with ConA. ConA does not bind mucin-type O-glycans, and even though MUC1 is moderately N-glycosylated, the authors have not shown that ConA does bind to their antigen used as a control.*

Response: We thank the reviewer for this critical comment. In our new experiment, CA 125 containing high-mannose N-glycans was used as a control of soluble glycoproteins labeled with ConA^{36, 37}. As soluble CA 125 with small size (a few nm) can be adequately filtered, the fluorescence signal of ConA-labeled CA 125 was negligible measured by EVLET. Additionally, the spiking of CA 125 into HD plasma did not significantly affect the performance of EVLET in detecting EV glycans. We have clarified this point in the revised manuscript.

Results

Design of EVLET

We further examined the impact of soluble proteins such as CA 125 that contains high-mannose N-glycans^{36, 37} on EV glycan detection. As ConA-labeled CA 125 could be efficiently filtered out with negligible signal, the spiking of CA 125 (35 U mL⁻¹) into healthy donor (HD) plasma did not significantly increase the fluorescence intensity of HD plasma measured by EVLET (Supplementary Fig. 15). In comparison, the spiking of MDA-MB-231 EVs (1.4×10⁸ EVs per assay) led to a noticeable increase in fluorescence signal of HD plasma.

Supplementary Fig. 15 | Minimal effect of soluble protein CA 125 on EV glycan detection. Fluorescence images of 5 types of samples measured by EVLET using FITC-ConA. (1) HD plasma spiked with CA 125 (35 U mL⁻¹) and MDA-MB-231 EVs (1.4×10⁸ EVs per assay); (2) HD plasma spiked with MDA-MB-231 EVs (1.4×10⁸ EVs per assay); (3) HD plasma spiked with CA 125 (35 U mL⁻¹); (4) HD plasma alone; (5) CA 125 (35 U mL⁻¹). Scale bar, 50 μm. Quantification of fluorescence intensities of different samples after thermophoretic accumulation was shown in the right panel ($n = 3$, mean ± s.d.).

Comment 7: Line 383 “Although ultracentrifugation has been widely adopted as the gold standard for isolating EVs from plasma samples, it suffers from co-precipitation of protein aggregates with EVs and relatively low and unsteady recovery of EVs that substantially affect the downstream detection of EV glycan.” As mentioned earlier, ultracentrifugation is not the gold standard for clinical analysis of EVs.

Response: We appreciate the reviewer’s comment. In the revised manuscript, we have removed the “gold standard” when describing ultracentrifugation for clinical analysis of EVs.

Discussion

Although ultracentrifugation has been widely used for EV isolation, it suffers from coprecipitation of protein aggregates and relatively low recovery of EVs, affecting the downstream detection of EV glycans^{38,39}.

Comment 8: *The clinical cohort in Figure 6 is small. As a discovery cohort this could be sufficient. However, the authors do not include a standard confirmation cohort to validate their model.*

Response: We appreciate the reviewer for this insightful comment. Our discovery cohort included 64 clinical samples, the size of which was over 20 times larger than the number of features (the signal intensities of three lectins). The ratio was considered sufficiently large for training the LDA model. Following the reviewer's suggestion, an independent cohort was used to validate the model, and an overall accuracy of 91 % was obtained for classifying TNBC, other BC, and HD in this validation cohort.

Results

Establishing an EV glycan signature for TNBC detection

In an independent validation cohort containing 32 plasma samples (8 TNBC patients, 11 other BC patients, and 13 HDs), the TNBC^{EGD} signature showed excellent performance (91 % accuracy) for classifying TNBC, other BC, and HDs (Fig. 6 and Supplementary Table 5).

Fig. 6 | EV glycan signature for TNBC detection. **a**, Heatmap of EV glycan profiles in the training cohort (20 TNBC patients, 19 other BC patients, and 25 HDs) and the

validation cohort (8 TNBC patients, 11 other BC patients, and 13 HDs) as measured by EVLET using a panel of 3 lectins (ConA, WGA, and RCA I). **b**, The TNBC^{EGD} signature in differentiating TNBC, other BC subtypes, and HD for the training and validation cohorts. **c**, Confusion matrix indicating the performance of the TNBC^{EGD} signature for the training cohort. **d**, Confusion matrix indicating the performance of the TNBC^{EGD} signature for the validation cohort.

Supplementary Table 5 | Summary of validation cohort for TNBC diagnosis.

Characteristic	TNBC	Other BC subtypes	HD	Total
Total cases	8	11	13	32
Subtypes				
HR+ HER2-	–	5	–	5
HR+ HER2+	–	3	–	3
HR- HER2+	–	3	–	3
TNBC	8	–	–	8
Age (year)				
Median (range)	54 (28-70)	53 (39-68)	54 (43 – 65)	54 (28-70)
Stage				
I	0	1	–	1
II	2	0	–	2
III	0	1	–	1
IV	4	9	–	13
Unknown	2	0	–	2
Serum CA 15-3 (U mL⁻¹)				
Median (range)	8.0 (2.8 – 131.0)	15.1 (6.2 – 164.2)	–	7.3 (2.8 – 164.2)

Comment 9: *Flow cytometry on cells treated with cisplatin is not sufficiently explained. Do the authors gate on viable cells? I.e. was a viability dye used? Given this information is not in the methods it would appear that no viability dye is used. As a result, there could be a significant impact on the flow cytometry results due to dead cells.*

Response: We thank the reviewer for these detailed inquiries. Per the reviewer’s request, flow cytometry has been repeated using the live/dead dye (Fixable Viability Dye) for cisplatin-treated cells. Flow cytometry analysis showed that the decreased cell viability was associated with a gradual decrease in fluorescence intensities of ConA-,

WGA- or RCA I-labeled MDA-MB-231 cells (Supplementary Figs. 28 – 29). This result indicated that the glycan pattern of MDA-MB-231 cells reflected the drug efficacy.

Methods

Drug response of TNBC cells

For flow cytometry analysis, the cells were labeled with FITC-conjugated lectins ($5 \mu\text{g mL}^{-1}$ ConA, RCA I; $20 \mu\text{g mL}^{-1}$ WGA) for 1 h at 4°C , followed by labelling with live/dead dye (Fixable Viability Dye eFluor™ 660, 1/1000) for 30 min at 4°C . The labeled cells were washed by $1\times$ PBS (centrifugation at $500\times g$ for 5 min) twice and suspended in $800 \mu\text{L}$ $1\times$ PBS containing 2 % FBS prior to the flow cytometric characterization (ACEA NovoCyte, Agilent Technologies, USA). For each sample, $>3,000$ cells were analyzed. The EVs were labeled with FITC-conjugated lectins and analyzed using the established EVLET protocols.

Supplementary Fig. 28 | Gating strategy for flow cytometry analysis of lectin-labeled MDA-MB-231 cells after cisplatin treatment. a, Cells gated based on FSC-H vs SSC-H. **b-c**, Single cells gated based on FSC-A vs FSC-H (**b**), and SSC-A vs SSC-H (**c**).

H (c). d, Live cells further gated on the less stained population after labelling with live/dead dye.

Supplementary Fig. 29 | Flow cytometry analysis of lectin-labeled MDA-MB-231 cells after cisplatin treatment. Cell viabilities of 100 % (untreated), 75 %, 50 %, and 25 % were selected. The intensity was normalized against the untreated samples ($n = 3$, mean \pm s.d.).

Comment 10: *The authors claim the cohorts are aged matched with each cohort age as 52 (35 –80) 52 (35 – 70) 45 (23 – 64). There seems to be a significant difference in the healthy donors. Could the authors at a minimum provide a statistical test showing no difference.*

Response: We appreciate the reviewer’s important comment. The nonparametric Mann-Whitney test showed that there was no statistically significant difference of age across TNBC, other BC, and HD cohorts. This additional detail has been included in the revised manuscript.

Supplementary Fig. 21 | Comparison of age distribution across TNBC, other BC, and HD cohorts. No statistically significant difference was observed between different

cohorts.

Comment 11: *It is unclear what advantage the proposed thermophoretic enrichment provides. The authors do not include a control of measuring the fluorescence of the sample from the post-vibrating membrane filtration system, thereby confirming their assay is providing an improvement to sensitivity of detection.*

Response: We thank the reviewer for this crucial comment. The thermophoretic enrichment not only amplifies the fluorescence signal of lectin-labeled EVs, but also reduces the sample volume to 14 μL due to the implementation of microfluidic chamber. In response to the reviewer's suggestion, we performed a control of measuring the fluorescence of the sample from the post-vibrating membrane filtration system (VMF) by a microplate reader. The limit of detection (LoD) of this method was two orders of magnitude higher than that of thermophoretic enrichment, indicating the substantial improvement in detection sensitivity by our EVLET. These experimental data have now been included in the revised manuscript.

Results

Design of EVLET

The limit of detection (LoD) was 4.1×10^5 EVs by EVLET, 2 orders of magnitude lower than that of lectin-based ELISA (3.8×10^7 EVs) or direct fluorescence measurement by microplate reader (1.1×10^8 EVs, Fig. 4d).

Fig. 4 | d, Sensitivity of EVLET (red, the left axis), lectin-based ELISA (blue, the right axis), and direct fluorescence measurement by microplate reader (green, the right axis) for detecting glycans on MDA-MB-231 EVs using ConA ($n = 3$, mean \pm s.d.).

Comment 12: *The authors used an ELISA to compare their results in Figure 4 to show improved sensitivity. However, this ELISA uses CD63 as a pull down. The authors have not demonstrated that their EVs are positive for CD63 (or any conventional EV marker).*

As a result, it is not known if the ELISA would work. Furthermore, due to the heterogeneity of EVs, not all EVs are CD63+, and as a result, only a subset of EVs would be analyzed using ELISA. Therefore, the sensitivity comparison EVs using thermophoretic enrichment compared to CD63+ EVs using ELISA is not appropriate or a valid comparison.

Response: We appreciate the insightful comment from the reviewer. We agree with the reviewer that due to an inherent heterogeneity of EVs, only a subset of EVs would be analyzed by ELISA with CD63 as a pull down. To address this issue, we used a hydrophobic substrate to adsorb EVs for lectin-based ELISA. This method achieved a limit of detection (LoD) of 3.8×10^7 EVs, approximately 100-fold higher than that of EVLET assay. These results have been included in the revised manuscript.

Results

Design of EVLET

The limit of detection (LoD) was 4.1×10^5 EVs by EVLET, 2 orders of magnitude lower than that of lectin-based ELISA (3.8×10^7 EVs) or direct fluorescence measurement by microplate reader (1.1×10^8 EVs, Fig. 4d).

Fig. 4 | d, Sensitivity of EVLET (red, the left axis), lectin-based ELISA (blue, the right axis), and direct fluorescence measurement by microplate reader (green, the right axis) for detecting glycans on MDA-MB-231 EVs using ConA ($n = 3$, mean \pm s.d.).

Methods

Lectin-based ELISA

EVs ($10^6 \mu\text{L}^{-1}$, 100 μL) were adsorbed onto 96-well ELISA plates with hydrophobic substrate (Thermo Scientific, USA) for 2 h at room temperature. Following adsorption, the supernatant was discarded, and the wells were blocked using 100 μL of $1 \times$ PBS

containing 1 % BSA for 1 h at room temperature. After discarding the supernatant, the wells were rinsed 3 times with 100 μ L of 1 \times PBS containing 1 % BSA. Subsequently, 100 μ L of solution containing lectin (5 μ g mL⁻¹ FITC-ConA, 20 μ g mL⁻¹ FITC-WGA, or 5 μ g mL⁻¹ FITC-RCA I) and 1 % BSA was added, and the plate was incubated for 1 h at room temperature. After labeling, the supernatant was removed, and the wells were washed 5 times with 100 μ L 1 \times PBS containing 1 % BSA. Fluorescence detection was then performed using a microplate reader (Synergy H1, Biotek, USA) with an excitation at 490 nm and an emission at 525 nm.

To evaluate the sensitivity and linear range of lectin-based ELISA for detecting EV glycans, 100 μ L of MDA-MB-231 EVs with different concentrations (0, 3.13 \times 10⁵, 6.25 \times 10⁵, 1.25 \times 10⁶, and 2.5 \times 10⁶ μ L⁻¹) were loaded onto ELISA plates, and then detected by the above protocol.

Comment 13: *The discussion is just a reiteration of the results and shows little insight, and in general could be improved. This is also pertinent as to why EV glycosylation is important in predicting treatment outcomes in patients.*

Response: We greatly appreciate the reviewer's comment. Per the reviewer's request, we have added more discussions about the significance of EV glycosylation in predicting treatment outcomes for cancer patients in the revised manuscript as follows.

Discussion

Tumor-derived EVs in blood plasma have increasingly received attention due to their abundance, high stability, and rich molecular cargos inherited from parental cells. However, most EV-based liquid biopsy studies focus on the detection and analysis of proteins or nucleic acids carried by EVs, while the potential of EV glycans is rarely investigated. Cancer-associated glycans, including α -2,6 sialic acid residues, high mannose, and complex type *N*-glycans, are abundantly expressed on EVs^{40, 41}. EV glycan signature in cancer ascites has been used for prognosis stratification of patients with colorectal and gastric cancers^{42, 43}. Monitoring glycosylation alterations of EVs secreted by chemotherapy-induced senescent TNBC cells provides valuable insight into the prediction of drug resistance and treatment efficacy^{44, 45}. Heavy *N*-glycosylation of PD-L1 on melanoma EVs facilitates the recognition and deactivation of PD-1⁺ CD8⁺ T cells, playing a vital role in inducing an immunosuppressive tumor microenvironment^{46, 47}. Thus, EV glycans can be considered as a new hallmark of cancer, showing great potential for non-invasive cancer detection, monitoring, and prognosis.

Comment 14: *Line 558 "The samples at the microchamber center were locally heated*

by a 1480-nm laser (Changchun Laser Optoelectronics Technology, China) with a power of 150 mW for 10 min for thermophoretic enrichment of EVs.” Although the authors provide a model for the proposed mechanism in Figure 4a, there is insufficient evidence as to what temperature is achieved and how this drives size-dependent accumulation of particles around the bottom of the microchamber. What is the minimum/maximum size of particles, what temperature is required etc.

Response: We appreciate the reviewer’s insightful comment. We have conducted numerical simulations to characterize the temperature profile and convection flow within the microchamber induced by the 1480 nm laser irradiation (Fig. 4a). A peak temperature of 65 °C was achieved at the heated core, resulting in a radial temperature gradient towards the cooler surroundings of room temperature (25 °C). This temperature gradient triggered two effects: thermophoresis, which repelled particles from the heated core (proportional to a^2 , where a is the particle diameter), and a convective flow that circulated particles via a drag force proportional to a . The interplay between thermophoresis and convection flow led to the size-dependent accumulation of particles at the bottom of the chamber. Our nanoparticle tracking analysis (NTA) showed that most of MDA-MB-231 EVs had a broad size range of 50 – 300 nm, with a modal size of 100 nm. Theoretically, all EVs within this range can be accumulated, and the simulation result indicated an over 2500-fold enrichment of EVs (50 – 300 nm) at the center of the microchamber bottom after a 10-min laser irradiation (Fig. 4b). Importantly, the temperature at the microchamber bottom was kept below 28 °C (a temperature rise of < 3 °C), due to the adoption of a high-thermal-conductivity sapphire substrate. This minimal temperature rise has negligible impact on glycan analysis of accumulated EVs. Based on the Reviewer’s comment, we have added these data and descriptions in revised manuscript.

Results

Design of EVLET

For thermophoretic detection of EV glycans, lectin-labeled EVs were first isolated and purified by VMF, followed by loading into a small microchamber for a 10-min local laser irradiation (1480 nm, 150 mW). The laser-induced radial temperature gradient between the heated core (up to 65 °C) and cold surrounding (25 °C) manifested two primary effects, thermophoresis that drove EVs away from the heated core (proportional to a^2 , a is the particle diameter), and convection flow for circulating EVs by exerting a drag force proportional to a (Fig. 4a). The counterbalance between the two effects resulted in an over 2500-fold enrichment of MDA-MB-231 EVs (50 – 300

nm in size based on NTA characterization) at the chamber bottom, as indicated by simulation results in Fig. 4b. Notably, using a sapphire substrate with a high thermal conductivity ($35 \text{ W m}^{-1} \text{ K}^{-1}$), the temperature around the bottom of microchamber was below $28 \text{ }^\circ\text{C}$, mitigating the impact of heat on EV glycan analysis.

Fig 4. | **a**, Mechanism of size-dependent accumulation of EVs under the interplay of thermophoresis and convection induced by the laser irradiation-generated temperature gradient (∇T). **b**, Numerical simulation of the spatial distribution of EVs at different time points. Scale bar, $400 \mu\text{m}$.

Methods

Numerical simulation

The flow field, heat transfer, and EV transportation in the microchamber were obtained by solving their government equations using a finite element method, as follows:

$$\begin{aligned}
 \nabla \cdot (\rho \mathbf{u}) &= 0 \\
 \frac{\partial (\rho \mathbf{u})}{\partial t} + \nabla \cdot (\rho \mathbf{u} \mathbf{u}) &= -\nabla p + \nabla \cdot (\eta \nabla \mathbf{u}) - \rho g \alpha \Delta T \\
 \frac{\partial (\rho T)}{\partial t} + \nabla \cdot (\rho \mathbf{u} T) &= \nabla \cdot \left(\frac{\lambda}{c_p} \nabla T \right) + S \\
 \frac{\partial c}{\partial t} + \nabla \cdot (\mathbf{u} c) &= \nabla \cdot (-D \nabla c - S_T D c \nabla T)
 \end{aligned} \tag{1}$$

where \mathbf{u} is the flow velocity vector, ρ is the fluid density, p is the pressure, η is the dynamic viscosity, c is the EV concentration, $S_T = 0.03(a/100 \text{ nm})^2 \text{ K}^{-1}$ is the Soret coefficient of EVs with a diameter of a , and $D = kT/3\eta\pi a$ is the diffusion coefficient of EVs. All governing equations were solved by a finite element solver (Comsol, Femlab). Gravity force was applied to the fluid motion to enable the thermal convection induced by the thermal expansion of water. The expansion rate α was derived from the temperature dependence of ρ . The thermal conductivity λ of water, glass, and sapphire were 0.6 , 1.3 , and $35 \text{ W m}^{-1} \text{ K}^{-1}$, respectively. The heat capacity c_p of water, glass, and sapphire were 4200 , 761 , and $755 \text{ J Kg}^{-1} \text{ K}^{-1}$, respectively. For the 1480 nm laser, the power was 150 mW and the diameter of focused spot was $160 \mu\text{m}$. The effect of size distribution of EVs was considered in the simulation. The percentages of EVs with

different size ranges at a step of 20 nm were obtained from the NTA measurement of MDA-MB-231 EVs post VMF. The simulation was performed for each center size (50 nm, 70 nm, 90 nm, etc) and the enrichment ratio of total EVs was determined as the weighted sum of enrichment ratio of EVs with each center size (weights were their percentages). The simulation result indicated that EVs can be accumulated by over 2500 folds at the center of microchamber bottom within 10 min of laser irradiation.

References

1. Amor AJ, *et al.* Relationship between noninvasive scores of nonalcoholic fatty liver disease and nuclear magnetic resonance lipoprotein abnormalities: A focus on atherogenic dyslipidemia. *J Clin Lipidol* **11**, 551-561.e557 (2017).
2. Manninen SM, Lankinen MA, de Mello VD, Laaksonen DE, Schwab US, Erkkilä AT. Intake of Fatty Fish Alters the Size and the Concentration of Lipid Components of HDL Particles and Camelina Sativa Oil Decreases IDL Particle Concentration in Subjects with Impaired Glucose Metabolism. *Mol Nutr Food Res* **62**, 1701042 (2018).
3. César TB, Oliveira MRM, Mesquita CH, Maranhão RC. High Cholesterol Intake Modifies Chylomicron Metabolism in Normolipidemic Young Men. *J Nutr* **136**, 971-976 (2006).
4. Mahmood Hussain M, Kancha RK, Zhou Z, Luchoomun J, Zu H, Bakillah A. Chylomicron assembly and catabolism: role of apolipoproteins and receptors. *Biochim Biophys Acta-Lipids Lipid Metab* **1300**, 151-170 (1996).
5. Bermudez V, *et al.* Lipid profile reference intervals in individuals from Maracaibo, Venezuela: an insight from the Maracaibo City Metabolic Syndrome prevalence study. *Rev Latinoam Hipertens* **7**, 24-34 (2012).
6. Greten H. Untersuchungen zum Stoffwechsel menschlicher Chylomikronen. *Wien Klin Wochenschr* **52**, 947-955 (1974).
7. Li Z, *et al.* Cascaded microfluidic circuits for pulsatile filtration of extracellular vesicles from whole blood for early cancer diagnosis. *Sci Adv* **9**, eade2819 (2023).
8. Rontogianni S, *et al.* Proteomic profiling of extracellular vesicles allows for human breast cancer subtyping. *Commun Biol* **2**, 325 (2019).
9. Muhsin-Sharafaldine M-R, Saunderson SC, Dunn AC, Faed JM, Kleffmann T, McLellan AD. Procoagulant and immunogenic properties of melanoma exosomes, microvesicles and apoptotic vesicles. *Oncotarget* **7**, 56279–56294 (2016).
10. Osteikoetxea X, *et al.* Differential detergent sensitivity of extracellular vesicle subpopulations. *Org Biomol Chem* **13**, 9775-9782 (2015).
11. Lane RE, Korbie D, Anderson W, Vaidyanathan R, Trau M. Analysis of exosome purification methods using a model liposome system and tunable-resistive pulse sensing. *Sci Rep* **5**, 7639 (2015).
12. Tutanov O, Proskura K, Kamyshinsky R, Shtam T, Tsentalovich Y, Tamkovich S. Proteomic Profiling of Plasma and Total Blood Exosomes in Breast Cancer: A Potential Role in Tumor Progression, Diagnosis, and Prognosis. *Front Oncol* **10**, 580891 (2020).
13. Arraud N, *et al.* Extracellular vesicles from blood plasma: determination of their morphology, size, phenotype and concentration. *J Thromb Haemost* **12**, 614-627 (2014).
14. Pasetto L, *et al.* Decoding distinctive features of plasma extracellular vesicles in amyotrophic lateral sclerosis. *Mol Neurodegener* **16**, 52 (2021).
15. Diehl JN, *et al.* A standardized method for plasma extracellular vesicle

- isolation and size distribution analysis. *PLoS One* **18**, e0284875 (2023).
16. Momen-Heravi F, *et al.* Impact of biofluid viscosity on size and sedimentation efficiency of the isolated microvesicles. *Front Physiol* **3**, 162 (2012).
 17. Baranyai T, *et al.* Isolation of Exosomes from Blood Plasma: Qualitative and Quantitative Comparison of Ultracentrifugation and Size Exclusion Chromatography Methods. *PLoS One* **10**, e0145686 (2015).
 18. Vogel R, *et al.* A standardized method to determine the concentration of extracellular vesicles using tunable resistive pulse sensing. *J Extracell Vesicles* **5**, 31242 (2016).
 19. Lobb RJ, *et al.* Optimized exosome isolation protocol for cell culture supernatant and human plasma. *J Extracell Vesicles* **4**, 27031 (2015).
 20. Han Z, *et al.* Highly efficient exosome purification from human plasma by tangential flow filtration based microfluidic chip. *Sens Actuators B: Chem* **333**, 129563 (2021).
 21. Hua X, *et al.* A double tangential flow filtration-based microfluidic device for highly efficient separation and enrichment of exosomes. *Anal Chim Acta* **1258**, 341160 (2023).
 22. Sunkara V, *et al.* Fully Automated, Label-Free Isolation of Extracellular Vesicles from Whole Blood for Cancer Diagnosis and Monitoring. *Theranostics* **9**, 1851-1863 (2019).
 23. Takov K, Yellon DM, Davidson SM. Comparison of small extracellular vesicles isolated from plasma by ultracentrifugation or size-exclusion chromatography: yield, purity and functional potential. *J Extracell Vesicles* **8**, 1560809 (2019).
 24. Wei R, *et al.* Combination of Size-Exclusion Chromatography and Ultracentrifugation Improves the Proteomic Profiling of Plasma-Derived Small Extracellular Vesicles. *Biol Proced Online* **22**, 12 (2020).
 25. Pang B, *et al.* Quality assessment and comparison of plasma-derived extracellular vesicles separated by three commercial kits for prostate cancer diagnosis. *Int J Nanomedicine* **15**, 10241-10256 (2020).
 26. Stranska R, *et al.* Comparison of membrane affinity-based method with size-exclusion chromatography for isolation of exosome-like vesicles from human plasma. *J Transl Med* **16**, 1 (2018).
 27. Pesce E, *et al.* Exosomes Recovered From the Plasma of COVID-19 Patients Expose SARS-CoV-2 Spike-Derived Fragments and Contribute to the Adaptive Immune Response. *Front Immunol* **12**, 785941 (2022).
 28. Buschmann D, *et al.* Evaluation of serum extracellular vesicle isolation methods for profiling miRNAs by next-generation sequencing. *J Extracell Vesicles* **7**, 1481321 (2018).
 29. Sidhom K, Obi PO, Saleem A. A Review of Exosomal Isolation Methods: Is Size Exclusion Chromatography the Best Option? *Int J Mol Sci* **21**, 6466 (2020).
 30. Chen Y, *et al.* Exosome detection via the ultrafast-isolation system: EXODUS. *Nat Methods* **18**, 212-218 (2021).

31. Visan KS, *et al.* Comparative analysis of tangential flow filtration and ultracentrifugation, both combined with subsequent size exclusion chromatography, for the isolation of small extracellular vesicles. *J Extracell Vesicles* **11**, 12266 (2022).
32. Veerman RE, *et al.* Molecular evaluation of five different isolation methods for extracellular vesicles reveals different clinical applicability and subcellular origin. *J Extracell Vesicles* **10**, e12128 (2021).
33. Maas SLN, *et al.* Possibilities and limitations of current technologies for quantification of biological extracellular vesicles and synthetic mimics. *J Control Release* **200**, 87-96 (2015).
34. Royo F, Théry C, Falcón-Pérez JM, Nieuwland R, Witwer KW. Methods for Separation and Characterization of Extracellular Vesicles: Results of a Worldwide Survey Performed by the ISEV Rigor and Standardization Subcommittee. *Cells* **9**, 1955 (2020).
35. Schuck P, Zhao H. The Role of Mass Transport Limitation and Surface Heterogeneity in the Biophysical Characterization of Macromolecular Binding Processes by SPR Biosensing. In: *Surface Plasmon Resonance: Methods and Protocols* (eds Mol NJ, Fischer MJE). Humana Press (2010).
36. Shang Y, Zeng Y, Zeng Y. Integrated Microfluidic Lectin Barcode Platform for High-Performance Focused Glycomic Profiling. *Sci Rep* **6**, 20297 (2016).
37. Kui Wong N, *et al.* Characterization of the Oligosaccharides Associated with the Human Ovarian Tumor Marker CA125. *J Biol Chem* **278**, 28619-28634 (2003).
38. Yan H, Li Y, Cheng S, Zeng Y. Advances in Analytical Technologies for Extracellular Vesicles. *Anal Chem* **93**, 4739-4774 (2021).
39. Shao H, Im H, Castro CM, Breakefield X, Weissleder R, Lee H. New Technologies for Analysis of Extracellular Vesicles. *Chem Rev* **118**, 1917-1950 (2018).
40. Batista BS, Eng WS, Pilobello KT, Hendricks-Muñoz KD, Mahal LK. Identification of a Conserved Glycan Signature for Microvesicles. *J Proteome Res* **10**, 4624-4633 (2011).
41. Krishnamoorthy L, Bess JW, Preston AB, Nagashima K, Mahal LK. HIV-1 and microvesicles from T cells share a common glycome, arguing for a common origin. *Nat Chem Biol* **5**, 244-250 (2009).
42. Wang Z, *et al.* Dual-Selective Magnetic Analysis of Extracellular Vesicle Glycans. *Matter* **2**, 150-166 (2020).
43. Wang Z, *et al.* Surfactant-guided spatial assembly of nano-architectures for molecular profiling of extracellular vesicles. *Nat Commun* **12**, 4039 (2021).
44. Kavanagh EL, *et al.* Protein and chemotherapy profiling of extracellular vesicles harvested from therapeutic induced senescent triple negative breast cancer cells. *Oncogenesis* **6**, e388-e388 (2017).
45. Kavanagh EL, *et al.* N-Linked glycosylation profiles of therapeutic induced senescent (TIS) triple negative breast cancer cells (TNBC) and their extracellular vesicle (EV) progeny. *Mol Omics* **17**, 72-85 (2021).

46. Marar C, Starich B, Wirtz D. Extracellular vesicles in immunomodulation and tumor progression. *Nat Immunol* **22**, 560-570 (2021).
47. Zhu L, *et al.* Coupling Aptamer-based Protein Tagging with Metabolic Glycan Labeling for In Situ Visualization and Biological Function Study of Exosomal Protein-Specific Glycosylation. *Angew Chem Int Edit* **60**, 18111-18115 (2021).

REVIEWER COMMENTS

Reviewer #1 (Remarks to the Author):

I am quite satisfied with the meritorious efforts to address my previous comments and would recommend the study to be published in its current form.

Reviewer #2 (Remarks to the Author):

The authors have addressed all comments except for one.

Comment 12: The authors used an ELISA to compare their results in Figure 4 to show improved sensitivity. However, this ELISA uses CD63 as a pull down. The authors have not demonstrated that their EVs are positive for CD63 (or any conventional EV marker). As a result, it is not known if the ELISA would work. Furthermore, due to the heterogeneity of EVs, not all EVs are CD63+, and as a result, only a subset of EVs would be analyzed using ELISA. Therefore, the sensitivity comparison EVs using thermophoretic enrichment compared to CD63+ EVs using ELISA is not appropriate or a valid comparison.

Response: We appreciate the insightful comment from the reviewer. We agree with the reviewer that due to an inherent heterogeneity of EVs, only a subset of EVs would be analyzed by ELISA with CD63 as a pull down. To address this issue, we used a hydrophobic substrate to adsorb EVs for lectin-based ELISA. This method achieved a limit of detection (LoD) of 3.8×10^7 EVs, approximately 100-fold higher than that of EVLET assay. These results have been included in the revised manuscript.

The issue I have with this is they used a 'hydrophobic' substrate to adsorb EVs to the surface. What is this hydrophobic substrate? And is that really suitable. Obviously EVs are a lipid rich object, but they are not hydrophobic.

Reviewer #3, expertise in glycan biology (Remarks to the Author):

The authors achieved rapid and high sensitivity for glycan profiling on extracellular vesicles by combining fluorescently labeled plant lectins with an ingenious ultrafiltration method.

Although glycan profiling using fluorescently labeled plant lectins is a widely used method, this study has the advantage of its rapid determination. In particular, good results have been obtained in the serodiagnosis of triple-negative breast cancer, although the number of samples is small.

Although this study is useful for its rapidity and sensitivity, some questions remain regarding the description of the study process, as follows;

Figure 3: Why does the study compare EV separation methods with various methods, but not tangential flow filtration (TFF)?

P9: The EV separation method is described as "VMF and SEC exhibited the high purity", but what does "purity" mean here?

p11: It is stated that 99.9% of FITC-labeled ConA is permeable, followed by "25.8 % of MDA-MB-231 EVs and 6.6 % of MCF-10A EVs" are labeled with this ConA, but I think this percentage is a different indicator. Also, since "DiD" is not defined, it is not clear what this experiment means, and therefore, the meaning of this sentence is not understood. Is it only the presence or absence of ConA labeling? Is the amount of labeling (fluorescence intensity) not subject to measurement?

P12, Figure S14: It is stated that the amount of ConA label is decreased by PNGaseF treatment, but the amount of change is small. Is there a large amount of mannose-presenting glycans other than N-glycan?

P12, Figure S15: Why was the uterine cancer marker (CA125) used for comparison? Wouldn't CA15-3 be a more appropriate comparison for breast cancer?

P12, Figure S16: Example of EV +, Plasma - is required.

P17~18, Fig. S27-30 "EV glycan signature for therapeutic response assessment": The text describes the fluorescence intensity of cells and EVs. Does this mean that only the absolute number of EVs changes with cell viability after cisplatin treatment, but not the glycan composition?

Authors' Response to the Review Comments

We are grateful to the reviewers for their constructive comments and insightful suggestions. Below, the reviewers' comments are shown in *blue*, and our responses are in black. Sections marked by **red** represent all altered contents in the revised manuscript. The following are our detailed responses.

Response to Reviewer #1

Comment 1: *I am quite satisfied with the meritorious efforts to address my previous comments and would recommend the study to be published in its current form.*

Response: We thank the reviewer for the positive evaluation of our research work, and the suggestion of acceptance of this work.

Response to Reviewer #2

Comment 1: *The authors have addressed all comments except for one.*

Comment 12: The authors used an ELISA to compare their results in Figure 4 to show improved sensitivity. However, this ELISA uses CD63 as a pull down. The authors have not demonstrated that their EVs are positive for CD63 (or any conventional EV marker). As a result, it is not known if the ELISA would work. Furthermore, due to the heterogeneity of EVs, not all EVs are CD63+, and as a result, only a subset of EVs would be analyzed using ELISA. Therefore, the sensitivity comparison EVs using thermophoretic enrichment compared to CD63+ EVs using ELISA is not appropriate or a valid comparison.

Response: We appreciate the insightful comment from the reviewer. We agree with the reviewer that due to an inherent heterogeneity of EVs, only a subset of EVs would be analyzed by ELISA with CD63 as a pull down. To address this issue, we used a hydrophobic substrate to adsorb EVs for lectin-based ELISA. This method achieved a limit of detection (LoD) of 3.8×10^7 EVs, approximately 100-fold higher than that of EVLET assay. These results have been included in the revised manuscript.

The issue I have with this is they used a 'hydrophobic' substrate to adsorb EVs to the surface. What is this hydrophobic substrate? And is that really suitable. Obviously EVs are a lipid rich object, but they are not hydrophobic.

Response: We appreciate the reviewer for this important comment. For passive immobilization of EVs, 96-well ELISA plates with polystyrene substrate (437112, Thermo Scientific, USA) were used in our work. The polystyrene microplate was commonly adopted for protein immobilization through hydrophobic interactions. Previous studies also confirmed that the polystyrene microplate could be used for unbiased EV capture regardless of the expression of tetraspanins (CD63, CD9, CD81) on EVs^{1,2}. To avoid confusion, we have removed the term “hydrophobic substrate” in the revised manuscript.

Methods

Lectin-based ELISA

EVs ($10^6 \mu\text{L}^{-1}$, 100 μL) were immobilized onto 96-well ELISA plates with polystyrene substrate (437112, Thermo Scientific, USA) through passive adsorption

for 2 h at room temperature^{1, 2}. After EV immobilization, the supernatant was discarded, and the wells were blocked using 100 μL of 1 \times PBS containing 1 % BSA for 1 h at room temperature. After discarding the supernatant, the wells were rinsed 3 times with 100 μL of 1 \times PBS containing 1 % BSA. Subsequently, 100 μL of solution containing lectin (5 $\mu\text{g mL}^{-1}$ FITC-ConA, 20 $\mu\text{g mL}^{-1}$ FITC-WGA or 5 $\mu\text{g mL}^{-1}$ FITC-RCA I) and 1 % BSA was added, and the plate was incubated for 1 h at room temperature. After labeling, the supernatant was removed, and the wells were washed 5 times with 100 μL 1 \times PBS containing 1 % BSA. Fluorescence detection was then performed using a microplate reader (Synergy H1, Biotek, USA) with an excitation at 490 nm and an emission at 525 nm.

Response to Reviewer #3

Comment 1: *The authors achieved rapid and high sensitivity for glycan profiling on extracellular vesicles by combining fluorescently labeled plant lectins with an ingenious ultrafiltration method.*

Although glycan profiling using fluorescently labeled plant lectins is a widely used method, this study has the advantage of its rapid determination. In particular, good results have been obtained in the serodiagnosis of triple-negative breast cancer, although the number of samples is small.

Although this study is useful for its rapidity and sensitivity, some questions remain regarding the description of the study process, as follows;

Response: We sincerely thank the reviewer for the positive comments and professional suggestions on our research work, which helped to improve the quality of our manuscript. The following comments have been thoroughly addressed upon the reviewer's request.

Comment 2: *Figure 3: Why does the study compare EV separation methods with various methods, but not tangential flow filtration (TFF)?*

Response: We thank the reviewer for this inquiry. We compared our vibrating membrane filtration (VMF) with ultracentrifugation (UC) and size exclusion chromatography (SEC), because UC and SEC are the two most commonly used EV isolation methods according to the International Society for Extracellular Vesicles (ISEV) survey³. Due to the limited access to tangential flow filtration (TFF), we did not experimentally compare our VMF with TFF. Alternatively, based on a comprehensive literature review regarding EV isolation by TFF, the comparison between VMF and TFF in terms of recovery rate, purify, sample volume and processing time has been summarized in Supplementary Table 3.

Supplementary Table 3 | Comparison between ultracentrifugation (UC), size exclusion chromatography (SEC), tangential flow filtration (TFF) and vibrating membrane filtration (VMF) in processing plasma/serum samples.

	UC	SEC	TFF	VMF (present study)
Recovery rate	1 % (present) < 5 % ^{4,5}	39.3 % (present) 60 % – 85 % ^{6,7}	60 % – 80 % ⁸⁻¹⁰	22.1 % (present)

Purity (particles/μg protein)	1.8×10^7 (present) $10^7 - 2 \times 10^8$ ^{8, 11, 12}	1.3×10^9 (present) $0.5 \times 10^9 - 6 \times 10^9$ ¹²⁻¹⁴	$10^7 - 1.2 \times 10^8$ ^{8, 10}	9.3×10^8 (present)
	Total protein: 99.3 %	Total protein: 99.4 %		Total protein: 99.5 %
Removal rate of contaminants	LDL: 96.7 % VLDL: 98.6 % (present)	LDL: 95.7 % VLDL: 88.7 % (present)	Total protein: 98.4 % ¹⁵	LDL: > 99.1 % VLDL: > 98.9 % (present)
Sample volume	2 mL (present) 1 mL – 4 mL ^{11, 12}	2 mL (present) 0.15 mL – 4 mL ^{11, 13, 16-18}	0.03 – 0.6 mL ⁸⁻¹⁰	0.002 mL (present)
Processing time	3 h (present) 1.4 – 3.5 h ^{11, 13, 19, 20}	3.5 h (present) 1.5 – 4 h ^{19, 21}	< 0.7 – 3 h ⁸⁻¹⁰	0.16 h (present)

Comment 3: P9: *The EV separation method is described as "VMF and SEC exhibited the high purity", but what does "purity" mean here?*

Response: We thank the reviewer for pointing this out. The purity is defined as the ratio of particle count to protein content (particles per μg protein) according to the Minimal information for studies of extracellular vesicles 2018 (MISEV-2018)²². To clarify, we have provided the definition of purity at its first appearance in the revised manuscript.

Results

Design of EVLET

In addition, a comprehensive list of assays suggested by the MISEV-2018 document was carried out for determining the recovery and purity (the ratio of particle count to protein content) of EVs isolated by VMF and other methods²².

Comment 4: p11: *It is stated that 99.9% of FITC-labeled ConA is permeable, followed by "25.8 % of MDA-MB-231 EVs and 6.6 % of MCF-10A EVs" are labeled with this ConA, but I think this percentage is a different indicator. Also, since "DiD" is not defined, it is not clear what this experiment means, and therefore, the meaning of this sentence is not understood. Is it only the presence or absence of ConA labeling? Is the amount of labeling (fluorescence intensity) not subject to measurement?*

Response: We appreciate the reviewer's important comment. The removal of unbound FITC-ConA by VMF was aimed to mitigate its impact on downstream thermophoretic analysis of EV glycan. We performed the fluorescence colocalization analysis of EVs labeled with FITC-ConA and DiD (a lipophilic dye for EV membranes) to determine the percentage of ConA-conjugated EVs from different cell lines. These clarifications have been made in the revised manuscript. Following the reviewer's suggestion, we

also measured the fluorescence intensities of FITC-ConA on EVs. The results showed that MDA-MB-231 EVs exhibited a higher averaged FITC intensity than MCF-10A EVs. These investigations have been added in the revised manuscript as follows.

Results

Design of EVLET

As unbound lectins may affect the detection accuracy, we first assessed the capability of VMF to remove free FITC-ConA without the presence of EVs. Due to its small size, 99.9 % of FITC-ConA could be filtered out by VMF (Supplementary Fig. 13). To determine the percentage of lectin-conjugated EVs derived from different cell lines, we used FITC-ConA to label EV glycans and DiD (a lipophilic dye for EV membranes) to stain all EVs. Fluorescence analysis revealed that 25.8 % of MDA-MB-231 EVs (TNBC cell line-derived EVs) and 6.6 % of MCF-10A EVs (benign breast cell line-derived EVs) had DiD-FITC colocalization. In addition, MDA-MB-231 EVs exhibited a higher averaged FITC intensity than MCF-10A EVs (1.2 folds, $p = 0.035$) (Supplementary Fig. 14). These results suggested an elevated expression of glycans on TNBC-derived EVs.

Methods

Reagents and materials

DiD (DiI18(5); 1,1'-dioctadecyl-3,3,3',3'-tetramethylindodicarbocyanine, 4-chlorobenzenesulfonate salt) was purchased from Thermo Fisher Scientific (USA).

Fluorescence colocalization analysis of EVs

To determine the percentage of lectin-conjugated EVs derived from different cell lines, FITC-ConA ($5 \mu\text{g mL}^{-1}$) was used to label MDA-MB-231 EVs and MCF-10A EVs ($10^7 \mu\text{L}^{-1}$, $200 \mu\text{L}$) at room temperature for 1 h. The EV samples were then filtered through VMF (6 kHz, 25 V, 50 kPa), washed twice with $150 \mu\text{L}$ PBS, and retrieved in $100 \mu\text{L}$ PBS. The recovered samples were treated with 4 nM lipophilic DiD to label all EVs at 37°C for 20 min. The fluorescence colocalization of DiD and FITC signals was analyzed under a fluorescence microscope (DMi8, Leica, Germany) with a $100\times$ objective, and the images were captured with an sCMOS camera (95B, Photometrics, Canada) using 2×2 -pixel binning and a 50 ms exposure time.

Supplementary Fig. 14 | Fluorescence colocalization of DiD- and Con A (FITC)-labeled EVs. a, MDA-MB-231 EVs and MCF-10A EVs subjected to ConA labeling, VMF and DiD labeling before fluorescence microscopy observation. The representative images are shown from three independent repeats. Scale bar, 20 μm . **b,** FITC fluorescence intensity of 90 DiD-FITC colocalization events for MDA-MB-231 EVs and 22 events for MCF-10A EVs. Statistical differences were determined by two-sided, nonparametric Mann–Whitney test (**b**). *P* value is indicated in the chart. Error bars represent the mean \pm s.d. in (**b**). Source data are provided as a Source Data file.

Comment 5: *P12, Figure S14: It is stated that the amount of ConA label is decreased by PNGaseF treatment, but the amount of change is small. is there a large amount of mannose-presenting glycans other than N-glycan?*

Response: We thank the reviewer for this important comment. Our data showed that the fluorescence intensity of ConA-labeled MDA-MB-231 EVs was reduced by 36 % after PNGase F treatment ($p = 0.0024$, Supplementary Fig. 15). The observation of partial decrease in ConA labeling post PNGase F treatment was also reported in previous studies^{23, 24}. We reasoned that the incomplete removal of mannose by PNGase F can be attributed to two factors: first, in addition to *N*-glycans, *O*-glycans also contain mannose²⁵, which cannot be cleaved by *N*-glycan-specific PNGase F. Second, EVs were treated by PNGase F without EV lysis and protein denaturation, so that the cleavage efficiency of PNGase F for EV surface glycans might be compromised due to the steric hindrance²⁶. This clarification has been added in the revised manuscript as follows.

Results

Design of EVLET

After peptide-N-glycosidase F (PNGase F) treatment, the fluorescence intensity of

ConA-labeled MDA-MB-231 EVs was reduced by 36 % due to the cleavage of N-linked mannose ($p = 0.0024$, Supplementary Fig. 15).

Comment 6: *P12, Figure S15: Why was the uterine cancer marker (CA125) used for comparison? Wouldn't CA15-3 be a more appropriate comparison for breast cancer?*

Response: We appreciate reviewer for the valuable suggestion. We used soluble CA 125 as control, because it contains high-mannose *N*-glycans and can be effectively recognized by ConA^{27, 28}. Upon the reviewer's request, we have also used CA 15-3, one of the most widely used serum marker in patients with breast cancer, for comparison. Similar to CA 125, the fluorescence signal of ConA-labeled CA 15-3 was negligible as measured by EVLET. Additionally, the spiking of CA 15-3 into HD plasma did not significantly affect the performance of EVLET in detecting EV glycans. We have added these experiment results in the revised manuscript.

Results

Design of EVLET

We further examined the impact of soluble proteins such as CA 125 and CA 15-3 on EV glycan detection. As ConA-labeled CA 125 and CA 15-3 could be efficiently filtered out with negligible signal, the spiking of CA 125 (35 U mL^{-1}) or CA 15-3 (25 U mL^{-1}) into healthy donor (HD) plasma did not significantly increase the fluorescence intensity of HD plasma measured by EVLET (Supplementary Fig. 16).

Supplementary Fig. 16 | Minimal effect of soluble protein CA 125 or CA15-3 on EV glycan detection. **a**, Fluorescence images of 8 types of samples measured by EVLET using FITC-ConA. (1) HD plasma spiked with CA 125 (35 U mL^{-1}) and MDA-MB-231 EVs (1.4×10^8 EVs per assay); (2) HD plasma spiked with CA 15-3 (25 U mL^{-1}) and MDA-MB-231 EVs (1.4×10^8 EVs per assay); (3) HD plasma spiked with MDA-MB-231 EVs (1.4×10^8 EVs per assay); (4) HD plasma spiked with CA 125 (35 U mL^{-1}); (5) HD plasma spiked with CA 15-3 (25 U mL^{-1}); (6) HD plasma alone; (7) CA 125 (35 U mL^{-1}); (8) CA 15-3 (25 U mL^{-1}). The representative images are shown from three independent repeats. Scale bar, $50 \mu\text{m}$. **b**, Quantification of fluorescence intensities of different samples after thermophoretic accumulation ($n = 3$ samples for each condition). Error bars represent the mean \pm s.d. Source data are provided as a Source Data file.

Comment 7: P12, Figure S16: Example of EV +, Plasma - is required.

Response: We appreciate the reviewer for this important suggestion. We have performed EVLET assay for MDA-MB-231 EVs without the presence of plasma (EV+, plasma-) and included the result in the revised manuscript as follows.

Supplementary Fig. 17 | Minimal effect of lipoproteins on EV glycan detection. a, Fluorescence intensities of 9 types of samples measured by EVLET: (1) HD plasma spiked with MDA-MB-231 EVs (1.4×10^8 EVs per assay) and LDL (2.7 mg mL^{-1}); (2) HD plasma spiked with MDA-MB-231 EVs and CM (1.3 mg mL^{-1}); (3) HD plasma spiked with MDA-MB-231 EVs; (4) MDA-MB-231 EVs; (5) HD plasma spiked with LDL; (6) HD plasma spiked with CM; (7) HD plasma; (8) LDL; (9) CM ($n = 3$ samples for each condition). **b,** Quantification of fluorescence intensities of different samples after thermophoretic accumulation ($n = 3$ samples for each condition). Error bars represent the mean \pm s.d. Source data are provided as a Source Data file.

Comment 8: P17~18, Fig. S27-30 "EV glycan signature for therapeutic response assessment": The text describes the fluorescence intensity of cells and EVs. Does this mean that only the absolute number of EVs changes with cell viability after cisplatin treatment, but not the glycan composition?

Response: We thank the reviewer for the insightful comment. To mitigate the potential bias due to the change of EV number, we first measured the concentration of EVs derived from cisplatin-treated cells by NTA and used the same amount of EVs

for EVLET. This ensures that the change in fluorescence intensity of EVs was attributed to the alternation in EV glycan composition, rather than fluctuation in EV number. We have clarified this point in the revised manuscript.

Results

EV glycan signature for therapeutic response assessment

We further investigated whether EV glycan profiles could be employed for assessing therapeutic response in TNBC. Cisplatin, one of the most commonly used chemotherapy drugs for TNBC, was selected. Cisplatin treatment of MDA-MB-231 cells engendered a dose-dependent decrease in the cell viability, and the IC₅₀ value (half-maximal inhibitory concentration) of cisplatin was determined as 80 μM for MDA-MB-231 cells (Supplementary Fig. 28). Moreover, the decreased cell viability after cisplatin treatment was correlated with declined fluorescence intensities of ConA-, WGA-, or RCA I-labeled MDA-MB-231 cells measured by flow cytometry (Supplementary Figs. 29 – 30). **The levels of ConA-, WGA- and RCA I-conjugated MDA-MB-231 EVs secreted by cisplatin-treated cells (unified to equal EV concentration) were also decreased as detected by EVLET (Supplementary Fig. 31).** We observed a strong correlation in fluorescence intensities between lectin-labeled MDA-MB-231 cells and the secreted EVs (Supplementary Fig. 31c), suggesting that EV glycan profiles could be considered as a potent indicator of drug efficacy.

Methods

Drug response of TNBC cells

For EVLET, the EVs secreted by cisplatin-treated cells were prepared at equal concentration ($5 \times 10^6 \mu\text{L}^{-1}$, 200 μL) and labeled with FITC-conjugated lectins ($5 \mu\text{g mL}^{-1}$ ConA, RCA I; $20 \mu\text{g mL}^{-1}$ WGA) for 1 h at room temperature. Then the samples were detected following the established EVLET protocol.

References

1. Jo A, *et al.* Inaugurating High-Throughput Profiling of Extracellular Vesicles for Earlier Ovarian Cancer Detection. *Adv Sci* **10**, 2301930 (2023).
2. Małys MS, Aigner C, Schulz SM, Schachner H, Rees AJ, Kain R. Isolation of Small Extracellular Vesicles from Human Sera. *Int J Mol Sci* **22**, 4653 (2021).
3. Royo F, Théry C, Falcón-Pérez JM, Nieuwland R, Witwer KW. Methods for Separation and Characterization of Extracellular Vesicles: Results of a Worldwide Survey Performed by the ISEV Rigor and Standardization Subcommittee. *Cells* **9**, 1955 (2020).

4. Momen-Heravi F, *et al.* Impact of biofluid viscosity on size and sedimentation efficiency of the isolated microvesicles. *Front Physiol* **3**, 162 (2012).
5. Baranyai T, *et al.* Isolation of Exosomes from Blood Plasma: Qualitative and Quantitative Comparison of Ultracentrifugation and Size Exclusion Chromatography Methods. *PLoS One* **10**, e0145686 (2015).
6. Vogel R, *et al.* A standardized method to determine the concentration of extracellular vesicles using tunable resistive pulse sensing. *J Extracell Vesicles* **5**, 31242 (2016).
7. Lobb RJ, *et al.* Optimized exosome isolation protocol for cell culture supernatant and human plasma. *J Extracell Vesicles* **4**, 27031 (2015).
8. Han Z, *et al.* Highly efficient exosome purification from human plasma by tangential flow filtration based microfluidic chip. *Sens Actuators B: Chem* **333**, 129563 (2021).
9. Hua X, *et al.* A double tangential flow filtration-based microfluidic device for highly efficient separation and enrichment of exosomes. *Anal Chim Acta* **1258**, 341160 (2023).
10. Sunkara V, *et al.* Fully Automated, Label-Free Isolation of Extracellular Vesicles from Whole Blood for Cancer Diagnosis and Monitoring. *Theranostics* **9**, 1851-1863 (2019).
11. Takov K, Yellon DM, Davidson SM. Comparison of small extracellular vesicles isolated from plasma by ultracentrifugation or size-exclusion chromatography: yield, purity and functional potential. *J Extracell Vesicles* **8**, 1560809 (2019).
12. Wei R, *et al.* Combination of Size-Exclusion Chromatography and Ultracentrifugation Improves the Proteomic Profiling of Plasma-Derived Small Extracellular Vesicles. *Biol Proced Online* **22**, 12 (2020).
13. Pang B, *et al.* Quality assessment and comparison of plasma-derived extracellular vesicles separated by three commercial kits for prostate cancer diagnosis. *Int J Nanomedicine* **15**, 10241-10256 (2020).
14. Stranska R, *et al.* Comparison of membrane affinity-based method with size-exclusion chromatography for isolation of exosome-like vesicles from human plasma. *J Transl Med* **16**, 1 (2018).
15. Dong L, *et al.* Comprehensive evaluation of methods for small extracellular vesicles separation from human plasma, urine and cell culture medium. *J Extracell Vesicles* **10**, e12044 (2020).
16. Pesce E, *et al.* Exosomes Recovered From the Plasma of COVID-19 Patients Expose SARS-CoV-2 Spike-Derived Fragments and Contribute to the Adaptive Immune Response. *Front Immunol* **12**, 785941 (2022).
17. Buschmann D, *et al.* Evaluation of serum extracellular vesicle isolation methods for profiling miRNAs by next-generation sequencing. *J Extracell Vesicles* **7**, 1481321 (2018).
18. Sidhom K, Obi PO, Saleem A. A Review of Exosomal Isolation Methods: Is Size Exclusion Chromatography the Best Option? *Int J Mol Sci* **21**, 6466 (2020).

19. Chen Y, *et al.* Exosome detection via the ultrafast-isolation system: EXODUS. *Nat Methods* **18**, 212-218 (2021).
20. Visan KS, *et al.* Comparative analysis of tangential flow filtration and ultracentrifugation, both combined with subsequent size exclusion chromatography, for the isolation of small extracellular vesicles. *J Extracell Vesicles* **11**, 12266 (2022).
21. Veerman RE, *et al.* Molecular evaluation of five different isolation methods for extracellular vesicles reveals different clinical applicability and subcellular origin. *J Extracell Vesicles* **10**, e12128 (2021).
22. Théry C, *et al.* Minimal information for studies of extracellular vesicles 2018 (MISEV2018): a position statement of the International Society for Extracellular Vesicles and update of the MISEV2014 guidelines. *J Extracell Vesicles* **7**, 1535750 (2018).
23. Taniguchi T, *et al.* N-Glycosylation affects the stability and barrier function of the MUC16 mucin. *J Biol Chem* **292**, 11079-11090 (2017).
24. Hanus C, *et al.* Unconventional secretory processing diversifies neuronal ion channel properties. *eLife* **5**, e20609 (2016).
25. Pinho SS, Reis CA. Glycosylation in cancer: mechanisms and clinical implications. *Nat Rev Cancer* **15**, 540-555 (2015).
26. Wang T, Voglmeir J. PNGases as Valuable Tools in Glycoprotein Analysis. *Protein Pept Lett* **21**, 976-985 (2014).
27. Shang Y, Zeng Y, Zeng Y. Integrated Microfluidic Lectin Barcode Platform for High-Performance Focused Glycomic Profiling. *Sci Rep* **6**, 20297 (2016).
28. Kui Wong N, *et al.* Characterization of the Oligosaccharides Associated with the Human Ovarian Tumor Marker CA125. *J Biol Chem* **278**, 28619-28634 (2003).

REVIEWERS' COMMENTS

Reviewer #2 (Remarks to the Author):

The authors have addressed the concerns raised adequately.

Reviewer #3 (Remarks to the Author):

The manuscript has been properly responded to, addressed and revised. It is ready for publication as is.

Authors' Response to the Review Comments

We are grateful to the reviewers for their constructive comments and insightful suggestions. Below, the reviewers' comments are shown in *blue*, and our responses are in black. The following are our detailed responses.

Response to Reviewer #2

Comment 1: *The authors have addressed the concerns raised adequately.*

Response: We thank the reviewer for the positive evaluation of our research work, and the suggestion of acceptance of this work.

Response to Reviewer #3

Comment 1: *The manuscript has been properly responded to, addressed and revised. It is ready for publication as is.*

Response: We thank the reviewer for the positive evaluation of our research work, and the suggestion of acceptance of this work.